# Simulating intersection angles between conjugate faults in sea ice with different VP rheologies

Damien Ringeisen [1], Martin Losch [1], L. Bruno Tremblay [2], and Nils Hutter [1]

[1]Alfred-Wegener-Institut, Helmholtz-Zentrum für Polar und Meeresforschung (AWI), Bremerhaven, Germany
[2]Department of Atmospheric and Oceanic Sciences, McGill University, Montréal, Quebec, Canada

**Correspondence:** Damien Ringeisen (damien.ringeisen@awi.de)

**Abstract.** Recent high resolution pan-Arctic sea ice simulations show fracture patterns (Linear Kinematic Features or LKFs) that are typical of granular materials, but with wider fracture angles than those observed in high-resolution satellite images. Motivated by this, ice fracture is investigated in a simple uni-axial loading test using two different Viscous-Plastic (VP) rheologies: one with an elliptical yield curve and a normal flow rule, and one with a Coulombic yield curve and a normal flow rule that applies only to the elliptical cap. With the standard VP rheology, it is not possible to simulate fracture angles smaller than $30°$. Further, the standard VP-model is not consistent with the behaviour of granular material such as sea ice, because: (1) the fracture angle increases with ice shear strength; (2) the divergence along the fracture lines (or LKFs) is uniquely defined by the shear strength of the material with divergence for high shear strength and convergent with low shear strength; (3) the angle of fracture depends on the confining pressure with more convergence as the confining pressure increases. This behavior of the VP model is connected to the convexity of the yield curve together with use of a normal flow rule. In the Coulombic model, the angle of fracture is smaller ($\theta = 23°$) and grossly consistent with observations. The solution, however, is unstable when the compressive stress is too large because of non-differentiable corners between the straight limbs of the Coulombic yield curve and the elliptical cap. The results suggest that, although at first sight the large scale patterns of LKFs simulated with a VP sea ice model appear to be realistic, the elliptical yield curve with a normal flow rule is not consistent with the notion of sea ice as a pressure-sensitive and dilatant granular material.

## 1  Introduction

Sea ice is a granular material, that is, a material that is composed of ice floes of different sizes and shapes (Rothrock and Thorndike, 1984; Overland et al., 1998). In most large-scale models, sea ice is treated as a viscous-plastic continuum. It deforms plastically when the internal stress becomes critical in compression, shear, or tension; it deforms as a very viscous (creep) flow when the internal stress is relatively small (e.g., Hibler, 1979; Zhang and Hibler, 1997; Hunke and Dukowicz, 1997). The corresponding highly non-linear sea-ice momentum equations can be solved with modern numerical solvers to reproduce, in a qualitative way, observed linear patterns of sea ice deformation within reasonable computing time (Hutchings et al., 2004; Lemieux et al., 2010; Losch et al., 2010; Hutter et al., 2018a). These Linear Kinematic Features (LKFs) are places of large shear and divergence (Kwok, 2001). Leads that open along LKFs are responsible for an emergent anisotropy

of such models, affecting the subsequent dynamics, mass balance, and the heat and matter exchanges between the ocean, ice and atmosphere. It is therefore important to investigate whether sea-ice fracture is represented accurately in continuum sea ice models.

The sea ice dynamics are complicated because of sharp spatial changes in material properties associated with discontinuities (e.g. along sea ice leads or ridges) and heterogeneity (spatially varying ice thickness and concentration). The sea ice momentum equations are difficult to solve numerically because of the non-linear sea ice rheology. Since the first sea ice dynamics model, the Elastic-Plastic sea ice model based on data collected during the Arctic Ice Dynamics Joint Experiment (AIDJEX Coon et al., 1974), several approaches to modeling sea ice have been developed. Sea ice has been modeled as an incompressible fluid (Rothrock, 1975), a Viscous-Plastic (VP) material (Hibler, 1979), an Elastic-Viscous-Plastic (EVP) material (Hunke, 2001), a granular material (Tremblay and Mysak, 1997), an Elastic Anisotropic Plastic (EAP) medium (Wilchinsky and Feltham, 2006), an elastic-decohesive medium (Schreyer et al., 2006), an Elasto-Brittle (EB) material (Rampal et al., 2016) and a Maxwell(viscous)-Elastic-Brittle (MEB) material (Dansereau et al., 2016). The actual number of approaches to sea-ice modeling in the community, however, is much smaller. For example, 30 out of 33 global climate models in CMIP5 use some form of the standard VP rheology (Stroeve et al., 2014).

In spite of its success, the standard VP rheology is not undisputed. Coon et al. (2007) critically reviewed the assumptions behind current modeling practice since the original model of Coon et al. (1974), namely the zero-tensile strength (ice is a highly fractured material) and isotropy assumptions of the sea ice cover and the rheological model. Originally, Coon et al. (1974) assumed sea ice to have cracks in all directions, justifying isotropic ice properties and isotropic rheologies. The use of continuum models such as the standard VP model for high-resolution simulations (grid spacings of 1–10 km) is also debated since the grid size approaches a typical floe size and clearly violates the continuum assumption. For instance, recent high-resolution simulations using the VP model used spatial resolution of approximately 500 m for a regional domain (Wang et al., 2006) and 1 km for a pan-Arctic domain (Hutter et al., 2018a). It can be argued that if the mode of deformation of a single floe is similar to that of an aggregate of floes, a given rheology developed for a continuum can still be applicable at spatial resolutions of the order of the floe size (Overland et al., 1998; Feltham, 2008, Appendix C), but the validity of a given flow rule across scales is not clear. At any scale, the assumption of viscous creep for small deformations is not physical and an elastic model would be appropriate for low stress states. The long viscous time scale, compared to the synoptic time scale of LKFs, of order 30 years (Hibler, 1979), however, allows viscous deformation to be viewed as a small numerical regularization with little implications for the dissipation of mechanical energy from the wind or ocean current (Bouchat and Tremblay, 2014), and the ice model can be considered as an ideal plastic material. Tsamados et al. (2013) included anisotropy explicitly in a VP model and show that it improved the representation of ice thickness and ice drift compared to an EVP model. Alternative VP rheologies were never widely used in the community. These include a Coulombic yield curve with a normal flow rule (Hibler and Schulson, 2000), a parabolic lens and a tear-drop (Pritchard, 1975), a diamond-shape yield curve with normal flow rules (Zhang and Rothrock, 2005), a Mohr-Coulomb yield curve with a double-sliding deformation law (Tremblay and Mysak, 1997) or a curved diamond (Wang, 2007).

Previously, fracture lines (LKFs) in the pack ice were explained by brittle fracture (Marko and Thomson, 1977). Similar fracture patterns were also observed, from the centimeter scale in the lab to hundreds of kilometers in satellite observations (Schulson, 2004; Weiss et al., 2007). The scale invariance of the fracture processes at the floe scale has not been shown. This may come from a lack of observations at both high spatial and temporal resolution. Based on satellite observations (e.g. RADARSAT Geophysical Processor System, RPGS, or Advanced Very-High-Resolution Radiometer, AVHRR), and in-situ internal ice stress measurements (e.g. from the Surface Heat Budget of the Arctic Ocean, SHEBA, experiment), Weiss et al. (2007) proposed to model winter sea ice as a material that undergoes brittle failure with subsequent inelastic deformation by sliding along LKFs. This idea was formalized with an additional parameterization to simulate damage associated with brittle fracture in an Elasto-Brittle (EB) and Maxwell-Elasto-Brittle (MEB) model (Girard et al., 2011; Rampal et al., 2016; Dansereau et al., 2016). We note that subsequent deformation in this model is considered as elastic deformation (EB) or visco-elastic deformation (MEB) instead of plastic. That is, in the EB and MEB approaches, the material does not weaken when fracture occurs, but rather the Young's modulus is reduced, leading to larger elastic deformation for the same stress. From the scaling behavior of simulated sea-ice deformation fields of EVP models (with 12 km grid spacing), it was found that the heterogeneity and the intermittency of deformation in the VP model are not consistent with RGPS data (Girard et al., 2009). In contrast, VP-models were shown to be indeed capable of simulating the PDFs of sea ice deformations and some of the scaling characteristics over the whole Arctic in agreement with the same observations, either with sufficient resolution (Spreen et al., 2017; Hutter et al., 2018a) or with tuned shear and compressive strength parameters (Bouchat and Tremblay, 2017).

High-resolution sea-ice models simulate LKF patterns in pack ice, where they appear as lines of high deformation (Hutchings et al., 2005; Hutter et al., 2018a). Previously fractured ice will be weaker and will affect future sea ice deformation fields. The weakening associated with shear deformation results from divergence and a reduction in ice concentration along the LKFs. This mechanism introduces an anisotropy in high resolution simulations that is similar to observations with comparable spatial resolution. Lead characteristics, including intersection angles between LKFs were studied a number of times (Lindsay and Rothrock, 1995; Hutchings et al., 2005; Wilchinsky et al., 2010; Bröhan and Kaleschke, 2014; Wang et al., 2016; Hutter et al., 2018b). These studies show that VP models produce LKFs with various confinements, scales, resolutions, and forcings. From observations with different instruments (Landsat, Seasat/SAR, areal photographs, AVHRR), typical fracture angles between intersecting LKFs of $(15 \pm 15)°$ emerge at scales from 1 km to 100 km (Erlingsson, 1988; Walter and Overland, 1993). Hutter et al. (2018b) present an LKF tracking algorithm and show that fracture angles (half of the intersection angles) between LKFs in RGPS data follow a broad distribution that peaks around $20°$, in line with previous assessments (e.g. Walter and Overland, 1993). Hutter et al. (2018b) also show that that the distribution of fracture angles in a VP simulation with 2-km grid spacing is biased with a high modal value of $45°$ and with too few small intersection angles between $15°$ and $25°$. The observed bias motivates the present investigation of the dependence of fracture angles in different VP rheologies and model settings, that is, scale, resolution, boundary conditions, model geometry, and variability in initial ice thickness field.

The simulation of fractures in sea ice models has been studied in idealized model geometries before. Hibler and Schulson (2000) investigated the effect of embedded flaws - that favors certain angles of fractures - in idealized experiments using a Coulombic yield curve. Hutchings et al. (2005) showed that LKFs can be simulated with an isotropic VP model using an

idealized model geometry. The shape of the elliptical yield curve (ratio of shear to compressive strength) in the standard VP model determines if ice arches can form in an idealized channel experiment (Hibler et al., 2006; Dumont et al., 2009). Pritchard (1988) investigated the yield curve's mathematical characteristics and derived angles between the principal stress directions and characteristics directions that depend on the tangent to the yield curve. These results show that stress states exist in plastic materials where no LKFs form and were later used to build a yield curve (Wang, 2007). To build an anisotropic rheology, Wilchinsky et al. (2010) used a Discrete Element Model (DEM) model in an idealized model domain and showed clear diamond-shaped fracture patterns. Idealized experiment are also used to investigate new rheologies, for example, the Maxwell-Elastic-Brittle (MEB) rheology (Dansereau et al., 2016) or the Material-Point Method (MPM) (Sulsky et al., 2007), or to study the theoretical framework explaining the fracture angles (e.g. Dansereau et al., 2017, with a Mohr-Coulomb yield criterion in an MEB model). Recently, Heorton et al. (2018) compared simulated fractures by the EVP and EAP models using an idealized model geometry and wind forcing, and showed that the anisotropic model creates sharper deformation features. To the best of our knowledge, the dependency of the fracture angles in sea ice on the shape of the yield curve using high resolution models has not yet been investigated. This is another motivation of this study.

In this paper, we simulate the creation of a pair of conjugate faults in an ice floe with two different VP rheologies in an idealized experiment at a spatial resolution of 25 m. We explore the influence of various parameters of the rheologies and the model geometry (Scale, resolution, confinement, boundary conditions, and heterogeneous initial conditions). The remainder of this paper is structured as follow : Section 2 presents the experimental setup: the VP framework (2.1), the definition of the yield curve (2.2), and the description of the idealized experiment (2.3). Section 3 presents the results: First the control simulation is presented (3.1), then we explore the sensitivity of the setup in section 3.2 to scale, resolution and longer run-time (3.2.1), modified boundary conditions and lateral confinement (3.2.2), and to heterogeneity in initial conditions (3.2.3). Finally, we consider the effects of two different yield curves with different flow rules in subsection 3.3: the elliptical (3.3.1) and the Coulombic yield curve (3.3.2). Discussion and conclusions follow in sections 4 and 5.

## 2 Experimental Setup

### 2.1 Viscous-Plastic model

We use the Massachusetts Institute of Technology general circulation model (MITgcm, Marshall et al., 1997) with its sea ice package that allows for the use of different rheologies (Losch et al., 2010). All thermodynamic processes have been turned off for our experiments. The initial sea ice conditions, mean (grid cell averaged) thickness $h$ and fractional sea ice cover $A$, are advected by ice drift velocities with a third order flux limiter advection scheme (Hundsdorfer et al., 1995). Ice drift is computed from the sea ice momentum equations

$$\rho h \frac{\partial \boldsymbol{u}}{\partial t} = -\rho h f \boldsymbol{k} \times \boldsymbol{u} + \boldsymbol{\tau}_{air} + \boldsymbol{\tau}_{ocean} - \rho h \nabla \phi(0) + \nabla \cdot \boldsymbol{\sigma}, \tag{1}$$

where $\rho$ is the ice density, $h$ is the grid cell averaged sea ice thickness, $\boldsymbol{u}$ is the velocity field, $f$ is the Coriolis parameter, $\boldsymbol{k}$ is the vertical unit vector, $\boldsymbol{\tau}_{air}$ is the surface air stress, $\boldsymbol{\tau}_{ocean}$ is the ocean drag, $\nabla \phi(0)$ is the gradient of sea surface height, and

$\sigma$ is the vertically integrated internal ice stress tensor. The form of $\sigma$ defines the rheology. In the case of the standard VP model described in (Hibler, 1979), the components of $\sigma$ are defined as

$$\sigma_{ij} = 2\eta_{ij}\dot{\varepsilon}_{ij} + (\zeta - \eta)\dot{\varepsilon}_{kk}\delta_{ij} - \frac{P}{2}\delta_{ij}, \tag{2}$$

where $\delta_{ij}$ is the Kronecker delta and summation over equal indices is implied. $\eta$ and $\zeta$ are the shear and bulk viscosities, $\dot{\epsilon}_{ij}$ is the strain rate tensor defined as

$$\dot{\epsilon}_{ij} = \frac{1}{2}\left(\frac{\partial u_i}{\partial x_j} + \frac{\partial u_j}{\partial x_i}\right), \tag{3}$$

and P is the maximum compressive stress defined as a function of the ice strength parameter $P^\star$, mean sea ice thickness $h$, and the sea ice concentration $A$:

$$P = P^\star h\, e^{-C^*(1-A)}, \tag{4}$$

where $C^\star$ is a free parameter.

The stress tensor $\sigma$ is often expressed in terms of principal stresses $\sigma_1$ and $\sigma_2$ or stress invariants $\sigma_I$ and $\sigma_{II}$. The principal stresses $\sigma_1$ and $\sigma_2$ are the principal components or eigenvalues of the stress tensor on a sea ice element. Eigenvalues always exist, because the stress tensor is by definition symmetric. The principal stresses $\sigma_1$ and $\sigma_2$ can be expressed as a function of $\sigma_{ij}$ as :

$$\sigma_1 = \frac{1}{2}\left(\sigma_{11} + \sigma_{22} + \sqrt{(\sigma_{11} - \sigma_{22})^2 + 4\sigma_{12}^2}\right), \tag{5}$$

$$\sigma_2 = \frac{1}{2}\left(\sigma_{11} + \sigma_{22} - \sqrt{(\sigma_{11} - \sigma_{22})^2 + 4\sigma_{12}^2}\right), \tag{6}$$

This change of coordinates can then be represented as a rotation of the coordinates by $\psi$ (Fig. A1). This angle is (Tremblay and Mysak, 1997):

$$\tan(2\psi) = \frac{2\sigma_{12}}{\sigma_{11} - \sigma_{22}}. \tag{7}$$

Any linear combination of the principal stresses are stress invariants. One common sets of stress invariants are the mean normal stress ($\sigma_I$) and the maximal shear stress ($\sigma_{II}$). They can be written as

$$\sigma_I = \frac{1}{2}(\sigma_1 + \sigma_2) = \frac{1}{2}(\sigma_{11} + \sigma_{22}), \tag{8}$$

$$\sigma_{II} = \frac{1}{2}(\sigma_1 - \sigma_2) = \frac{1}{2}\sqrt{(\sigma_{11} - \sigma_{22})^2 + 4\sigma_{12}^2}. \tag{9}$$

## 2.2 Yield curve

The VP rheology was originally developed to simulate ice motion on a basin scale (e.g., Arctic Ocean, Southern Ocean) (Hibler, 1979). In this model, stochastic elastic deformation is parameterized as highly viscous (creep) flow (Hibler, 1977). Ice is set

**Table 1.** Model parameters of the reference simulation

| Symbol | Definition | Value | Unit |
|---|---|---|---|
| $\rho$ | Density of ice | 910 | $\mathrm{kg\,m^{-3}}$ |
| $P^{\star}$ | Ice strength | 27.5 | $\mathrm{kN\,m^{-1}}$ |
| $C$ | Strength reduction parameter | 20 | |
| $\Delta_{min}$ | Maximum Viscosity | $10^{-10}$ | $s^{-1}$ |
| $\Delta x, \Delta y$ | Grid spacing | 25 | m |
| $C_w$ | Water drag coefficient | $5.21 \times 10^{-3}$ | |
| $\mathrm{N}_x, \mathrm{N}_y$ | Size of the domain | $400 \times 1000$ | |
| $\mathrm{L}_x, \mathrm{L}_y$ | Size of experiment | $10 \times 25$ | km |
| $\mathrm{l}_x, \mathrm{l}_y$ | Ice floe's size | $8 \times 25$ | km |
| $A$ | Initial ice concentration | 100 | % |
| $h$ | Initial ice thickness | 1.0 | m |
| $N_{lin}$ | Nbr. linear iteration | 1500 | |
| $N_{nlin}$ | Nbr. non-linear iteration | 1500 | |
| $\epsilon_{err}$ | Max. error in LSR | $10^{-11}$ | $\mathrm{m\,s^{-1}}$ |
| $dt$ | Timestep | 0.1 | s |
| $e$ | Ellipse ratio $(a/b)$ | 2.0 | |
| $v_i$ | Initial velocity | 0 | $\mathrm{m\,s^{-1}}$ |
| $a_v$ | Acceleration | $5 \cdot 10^{-4}$ | $\mathrm{m\,s^{-2}}$ |

in motion by surface air and basal ocean stresses moderated by internal ice stress. When the internal sea-ice stress reaches a critical value in compression, tension or shear, sea ice fails and relatively large plastic deformation takes place. Internal ice stress below these thresholds leads to highly viscous (creep) flow that parameterizes the bulk effect of many small reversible elastic deformation events. The timescale of viscous deformation is so high ($\simeq 30\,\mathrm{years}$) that viscous deformation can be seen
5    as regularisation for better numerical convergence in the case of small deformation. Plastic deformations are relatively large and irreversible. Viscous deformations are negligibly small; in contrast to elastic deformation, they are also irreversible. The yield criterion is expressed as a 2D envelope either in principal stress space or stress invariant space with a normal flow rule. The constitutive equations (Eq. 2) are derived assuming that the principal axes of stress coincide with the principal axes of strain. The stress state on the yield curve together with the normal flow rule therefore determines the relative importance of
10    divergence (positive or negative) and shear strain rate at a point. The magnitude of the deformation is such that the stress state remains on the yield cure during plastic deformation.

In this study, we use two different yield curves: an elliptical yield curve (Hibler, 1979) and a Coulombic yield curve (Hibler and Schulson, 2000). The elliptical yield curve is used in conjunction with a normal flow rule, while the Coulombic yield curve

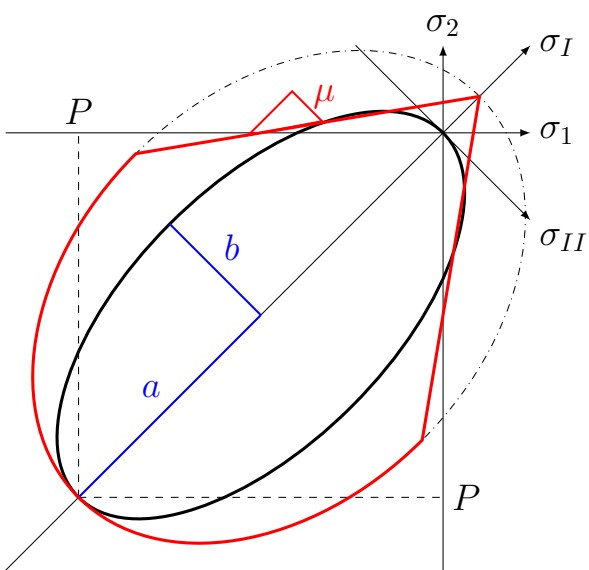

**Figure 1.** Elliptical yield curve (black) with ellipse aspect ratio $e = a/b = 2$. Coulombic yield curve (red) and elliptical capping with internal angle of friction ($\mu$). Both $e$ and $\mu$ are measures of the shear strength of the material. The normal flow rule applies only to the elliptical part of the yield curves. For the two straight limbs of the Coulombic yield curve, the flow is normal to the truncated ellipse (dash-dot line) with the same first stress invariant. Note that the axes $\sigma_1$, $\sigma_2$ and $\sigma_I$, $\sigma_{II}$ do not have the same scale.

uses a normal flow rule on the elliptical cap and a flow rule normal to the truncated ellipse for the same first principal stress (Hibler and Schulson, 2000, Appendix A). For the elliptical yield curve (Fig. 1, black line), $\eta$ and $\zeta$ are given by :

$$\zeta = \frac{P}{2\Delta}, \tag{10}$$

$$\eta = \frac{\zeta}{e^2}, \tag{11}$$

5 with the abbreviation

$$\Delta = \sqrt{\dot{\epsilon}_I^2 + \frac{1}{e^2}\dot{\epsilon}_{II}^2}. \tag{12}$$

In this abbreviation, the strain rate invariants are the divergence $\dot{\epsilon}_I = \dot{\epsilon}_{11} + \dot{\epsilon}_{22}$ and the maximum shear strain rate $\dot{\epsilon}_{II} = \sqrt{(\dot{\epsilon}_{22} - \dot{\epsilon}_{11})^2 + 4\dot{\epsilon}_{12}^2}$. $e = \frac{a}{b}$ is the ellipse aspect ratio with the semi-major half-axes $a$ and $b$ (shown in blue in Fig. 1). The ellipse aspect ratio $e$ defines the shear strength $S^\star = \frac{P^\star}{2e}$ of the material as a fraction of its compressive strength (Bouchat and

10 Tremblay, 2017). For the Coulombic yield curve (Fig. 1, red curve), the shear viscosity $\eta$ is capped on the two straight limbs:

$$\eta_{MC} = \min\left\{\eta, \frac{1}{\dot{\epsilon}_{II}}\left[\mu\left(\frac{P}{2} - \zeta \cdot \dot{\epsilon}_{kk}\right) - c\right]\right\} \tag{13}$$

where $\mu$ is the slope of the Mohr-Coulomb limbs (Fig. 1), $c$ is the cohesion value (the value of $\sigma_{II}$ for $\sigma_I = 0$) defined relative to the tensile strength by $c = \mu \cdot T^\star$, where $T^\star$ is defined as a fraction of $P^\star$.

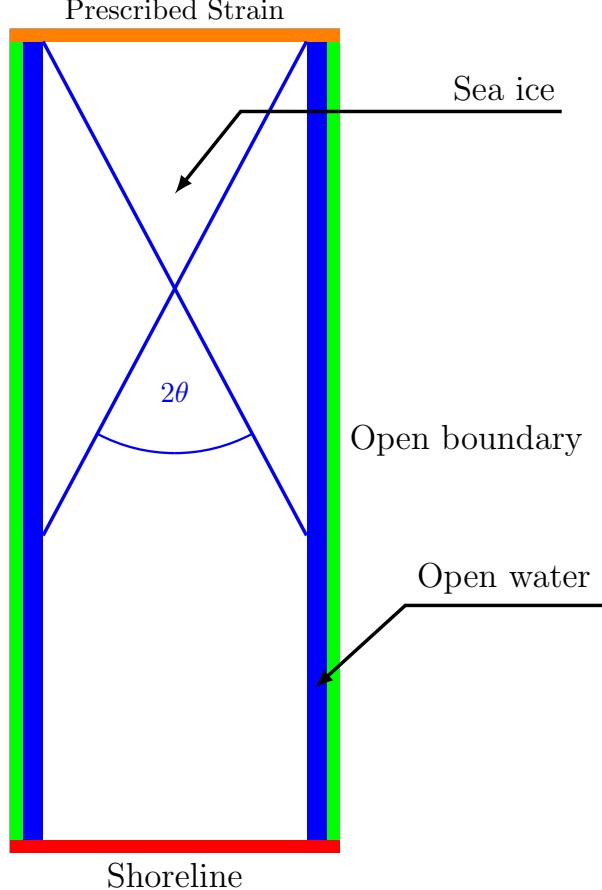

**Figure 2.** *Model domain with a solid wall on the southern (red) boundary (Dirichlet boundary conditions with $\boldsymbol{u} = 0$), and prescribed southward velocities on the northern orange boundary ($u = 0$, $v = a_v \cdot t + v_i$, Eq. 15) and open boundaries to the East and the West (green) with von Neumann boundary conditions. $\theta$ is the measured fracture angle with the blue line representing an LKF.*

The theoretical angle of fracture $\theta$ can be calculated from the Mohr's circle of stress and yield curve written in the local (reference) coordinate system (Ip et al., 1991; Pritchard, 1988; Hibler and Schulson, 2000). Details are described in the Appendix B. For a Mohr-Coulomb yield criterion, $\theta$ follows immediately from the internal angle of friction or the material shear strength. An instructive analogue is the slope of a pile of sand on a table. Moist sand has a higher shear strength and hence the slope angle can be steeper (i.e. the angle $\theta$ is smaller).

### 2.3 Idealized Experiment

An idealized compressive test is used to investigate the modes of sea ice fracture (Figure 2). This experiment is standard in engineering (Schulson, 2004; Weiss et al., 2007). The numerical configuration is inspired by Herman (2016) and similar to the

one shown in Dansereau et al. (2016). All experiments presented below use the same set-up unless specified otherwise. The values of parameters and constants are presented in Table 1.

The model domain is a rectangle of size $10\,\text{km} \times 25\,\text{km}$, except for the experiments presented in Sect. 3.2.1 and Sect. 3.2.2. An ice floe of size $8\,\text{km} \times 25\,\text{km}$, surrounded by $1\,\text{km}$ of open water on the eastern and western sides, is compressed with a linearly (in time) increasing strain rate from the North against a solid southern boundary. The eastern and western strips of open-water avoid interesting dynamics to be confounded by the choice of lateral boundary conditions along the open boundaries. We use a no-slip condition for the southern boundary, constraining lateral ice motion. Note that the results presented below are not sensitive to the choice of boundary condition on the eastern and western boundaries. Because the simulation time and the ice velocities are small, the Coriolis force in the momentum equations are neglected. Ocean and sea ice are initially at rest. The only term left in the momentum equation (Eguation (1) that is relevant for our experiment is the stress divergence term, $\nabla \cdot \sigma$. The ice floe has a uniform concentration of 100% and a thickness of $1\,\text{m}$. The spatial resolution of the model is $25\,\text{m}$. The angle of fracture is measured with the angle measuring tool of the GNU Image Manipulation Program (GIMP, https://www.gimp.org/). All angles measured in this study have an error of approximately $1°$. The finite size of the grid spacing widens the deformation line, and the fracture spreads over several pixels because of the obliquity of the fracture. Automatic algorithms for measuring LKF intersection angles are described in Linow and Dierking (2017); Hutter et al. (2018b).

We solve the non-linear sea-ice momentum equations with a Picard or fixed point iteration with 1500 non-linear or outer-loop (OL) iterations. Within each non-linear iteration, the non-linear coefficients (drag coefficients and viscosities) are updated and a linearized system of equations is solved with a Line Successive (over-)Relaxation (LSR) (Zhang and Hibler, 1997). The linear iteration is stopped when the maximum norm of the updates is less than $\epsilon_{LSR} = 10^{-11}\,\text{m}\,\text{s}^{-1}$, but we also limit the number iterations to 1500. Typically, 1500 non-linear iterations are required to reach a state close enough to the converged solution. Note that this criterion is much stricter than that proposed by (Lemieux and Tremblay, 2009) — this is so because of slow convergence due to the highly non-linear rheology term and the high spatial resolution.

On the open eastern and western boundaries, we use von Neumann boundary conditions for velocity, thickness and concentration and ice can escape the domain without any restrictions:

$$\left.\frac{\partial u}{\partial x}\right|_{E,W} = \left.\frac{\partial v}{\partial x}\right|_{E,W} = \left.\frac{\partial A}{\partial x}\right|_{E,W} = \left.\frac{\partial h}{\partial x}\right|_{E,W} = 0, \tag{14}$$

where $E$ and $W$ denote the eastern and western boundaries, respectively. Strain is applied to the ice at the northern boundary by prescribing a velocity that increases linearly with time :

$$v|_N(t) = a_v \cdot t + v_i \ ; \ u|_N = 0 \ ; \ \left.\frac{\partial A}{\partial y}\right|_N = \left.\frac{\partial h}{\partial y}\right|_N = 0, \tag{15}$$

where $a_v$ is the prescribed acceleration, and N denotes the northern boundary.

## 3   Results

We use simple uni-axial loading experiments to investigate the creation of pair of conjugate faults and their intersection angle. After presenting the results of simulations with the default parameters (Section 3.1), we explore the effects of experimental

choices: confining pressure, choice of boundary conditions (i.e. von Neumann versus Dirichlet), domain size and spatial resolution and inhomogeneities (i.e. localized weakness) in the initial thickness and concentration field (Section 3.2). Finally, we study the behaviour of two viscous plastic rheologies with different yield curves and compare these dependencies to what we can infer from smaller and larger scale measurements from laboratory experiment and RGPS observations (Section 3.3).

## 3.1 Uni-axial compressive test - Default parameters

With default parameters (Table 1), a diamond shape fracture appears in the shear strain rate and divergence fields after a few seconds of integration (Figure 3). After 1 timestep (or $0.1\,\text{s}$), the stress states already lie on the yield curve and the fracture is readily seen in the deformation fields (divergence and shear). We iterate for a total of 20 seconds in order for the signal to be apparent in the thickness and concentration fields. We do this to more clearly show the link between position of the stress states on the yield curve and the resulting deformation defined by the normal flow rule in the standard VP rheology of Hibler (1979). The shear deformation ($\dot{\epsilon}_{II}$) shows where the ice slides in friction and deforms plastically. From Fig. 3, the simulated intersection angle is $\theta = (34 \pm 1)°$.

After a few time steps, the ice thickness decreases particularly along the LKFs (Fig. 3c) where divergence is maximal. Note that the loading axis in our simple 1D experiment is also the second principal axis and consequently the stress states are migrating along the $\sigma_2$ axis as the strain rate at the northern boundary increases. Fracture occurs after plastic failure when the stress state reaches the yield curve and the ice starts to move in divergence. This occurs in the half of the ellipse closer to the origin (for $e > 1$) where the normal to the flow rule points in the direction of positive divergence (or first strain rate invariant) (see Fig. 4). This explains the simulated divergent flow field and lower ice thickness particularly along LKFs.

## 3.2 Sensitivity experiments

In this section, we test the sensitivity of the standard VP model simulation (Sect. 3.1) to the choice of resolution, scale, and run-time (3.2.1), boundary conditions and confinement pressure (3.2.2), and heterogeneity in the initial sea ice mass field (3.2.3).

### 3.2.1 Domain size, spatial resolution and length of integration

The angle of intersection between a pair of conjugate faults does not change with domain size and spatial resolution (Fig. 5). This is expected, because non-dimensionalizing the divergence of the internal ice stress term (the only term that remains in this simple uni-axial test experiment) by setting $u' = u/U$, $x' = x/L$, gives the same equations in non-dimensional form irrespective of the initial ice thickness or spatial resolution. Consequently, the control and sensitivity experiments are scale independent and the behaviour of the standard VP model can be readily compared with results from RGPS, AVHRR, or laboratory experiments.

Continuing the integration to 2700 seconds (45 min), compared to 20 seconds in the reference simulation leads to the creation of smaller diamond-shaped ice floes due to secondary and tertiary fracture lines (Figure 6). The openings are visible in the

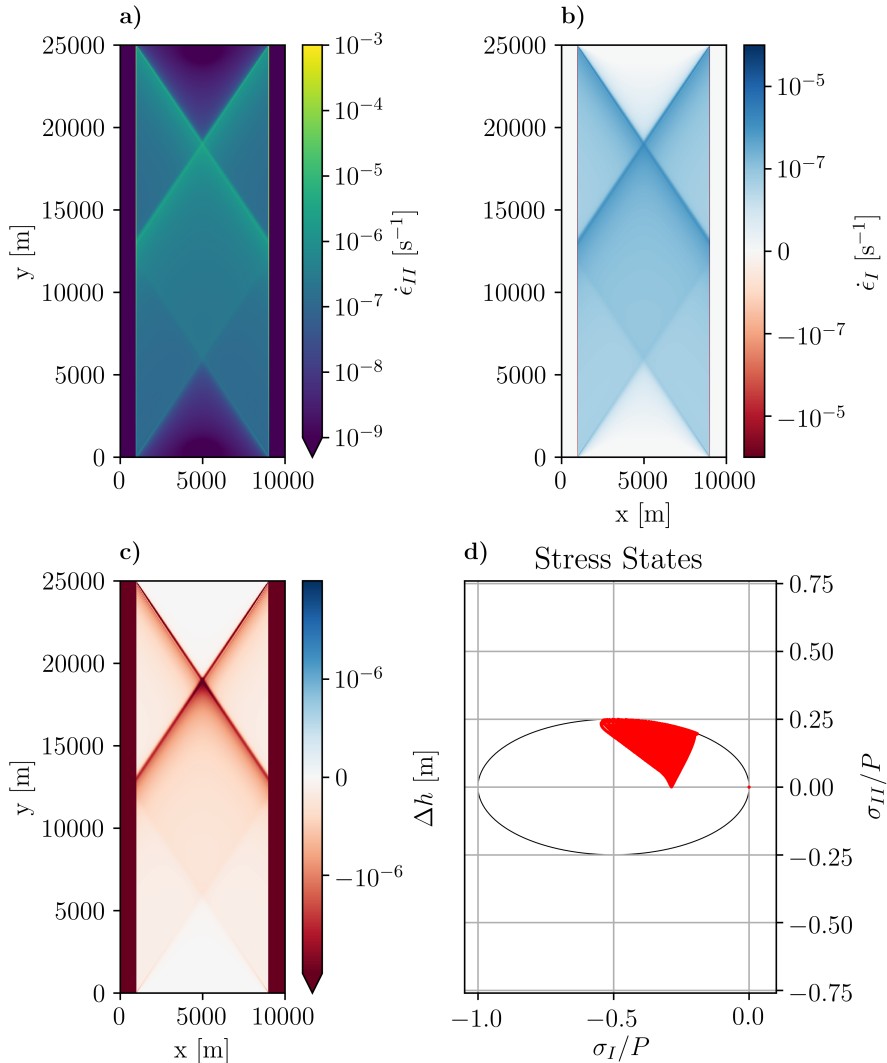

**Figure 3.** (a) First and (b) second strain invariants, (c) ice thickness anomaly ($\Delta h = h - 1$) and (d) stress states in normalized stress invariant space along with the elliptical yield curve after 5 seconds of integration. The first and second strain invariants represent the divergence and maximum shear strain rate, respectively. The modeled angle of fracture is $\theta = (34 \pm 1)\,^\circ$

thickness and concentration fields with thinner, less concentrated ice in the lead. In this longer experiment, the sea ice also ridges, for instance at the center of the domain where the apex of the diamonds fails in compression. There is also some thicker ice at the northern boundary induced by the specified strain rate at the northern boundary. The fracture pattern and presence of secondary and tertiary fracture lines are in line with results from laboratory experiments Schulson (2004) and with AVHRR and RGPS observations.

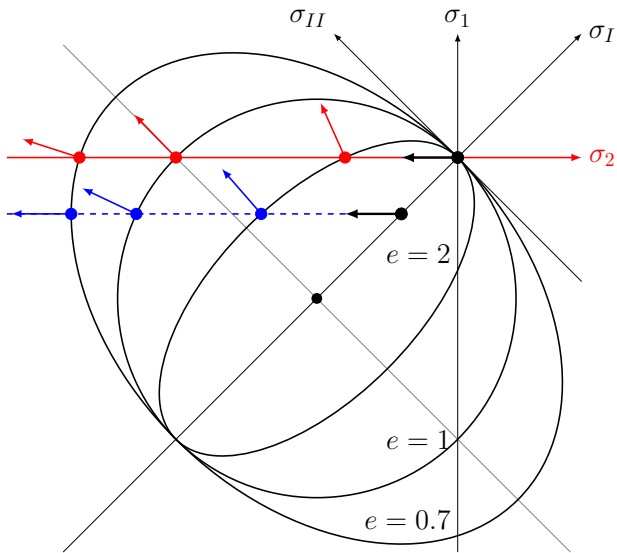

**Figure 4.** Schematic of stress states and failure in principal stress space. Black arrows show how stresses move from zero at the beginning of loading towards the yield curve until failure. Red points show the stress states at failure — the intersection point between the second principal axis 2 (in red) and the elliptical yield curve — for different ellipse ratios $e = 2, 1, 0.7$. The red arrows show the direction of deformation with a normal flow rule. The blue points and arrows show the case when the ice floe is confined and the loading will lead to extra stress in the direction of $\sigma_1$.

In the following, we always show results after 5 seconds of integration because our main focus is on the initial fracture of the ice, that is, the instant when the ice breaks for the first time under compression.

### 3.2.2 Boundary conditions and geometry

The dynamics responsible for the ice fracture and location of the fracture (presented above) take place far away from the
5  eastern and western boundaries and therefore do not depend on the choice of the corresponding boundary conditions. We now investigate the sensitivity of the results to the choice of boundary condition at the southern boundary. To this end, we force the fracture line to intersect the southern boundary by reducing the domain size to $10\,\mathrm{km} \times 10\,\mathrm{km}$ with an ice floe of $8\,\mathrm{km} \times 10\,\mathrm{km}$ in the interior. In this case, the fracture develops from corner to corner and the angle is solely determined by the geometry of the ice floe, that is, $\theta = \arctan(l_x/l_y)$ (Fig. 7b). With a free-slip boundary condition at the southern boundary, the fracture angle is
10  similar to the one from the control simulation (Fig. 7a). That is, the no-slip condition concentrates the stress to the corner of the ice floe touching the boundary and pre-determines the fracture location. A free-slip boundary condition is therefore considered more physical in such idealized experiments where fractures lines can extend from one boundary to another. This result can have implications for simulation of LKFs in the Arctic that would extend from one boundary to another, for instance in the Beaufort Sea.

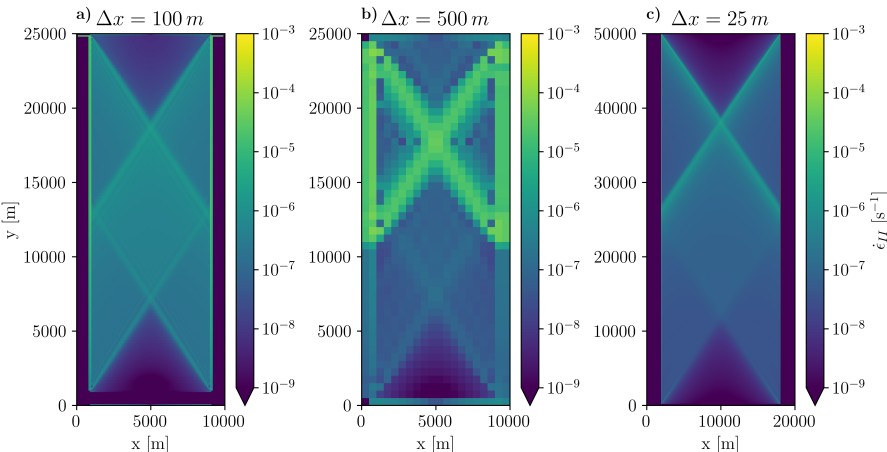

**Figure 5.** Maximum shear strain rate (second strain invariant) after 10 seconds of integration for the default domain size and $\Delta x = 100\,\mathrm{m}$ (a) and $500\,\mathrm{m}$ (b), and for the default $\Delta x$ and a doubled domain size of $20\,\mathrm{km} \times 50\,\mathrm{km}$ (c). Note that for case of the double domain (c), the southward velocity at the northern boundary was also doubled to keep the deformation rate constant, and that this simulation is limited to 2 seconds for numerical efficiency.

No-slip or free-slip boundary conditions have little impact on the fracture angle in the larger domain used in the control run simulation, because the LKFs always only touch one boundary and end in open-water (results not shown). With the free-slip boundary conditions, the stresses and strains are only different south of the diamond fracture pattern because ice can move along the southern boundary and the second fracture cannot form.

We now explore the effect of confining pressure on the eastern and western boundaries on the angle of fracture when using a (convex) elliptical yield curve with a normal flow rule. To do so, we replace the open boundaries to the East and the West with solid walls and the open water gaps with ice of thicknesses $h_c$. Note that the ice strength is linearly related to the ice thickness (Eq. 4). Therefore the normal stress at the edge of the floe is completely defined by the thickness of the surrounding ice.

With an increasing lateral confinement pressure (i.e. an increasing ice thickness $h_c$ next to the main floe), all stress states are moved to higher compressive stresses (blue curve in Fig. 4) and the fracture angle increases (Figure 8). In this case, the

stress states are again migrating in a direction parallel to the $\sigma_2$ axis but with a non-zero $\sigma_1$ value. The stress states of the ice along the fracture are therefore located in a region of higher compressive stresses on the yield curve where the divergence is reduced or even changes sign. With increasing confinement, the stress states of the ice floe move to more negative values of $\sigma_1$ along a line of constant $\sigma_2$ (blue line in Fig. 4) with deformation moving towards more convergent states. Between

$h_c = 0.2$ and $h_c = 0.3$, the regime changes from lead opening to ridging, as the fracture angle increases to values above $45°$. This is inconsistent with the behavior of a granular material where the angle of fracture is independent of confining pressure in uni-axial loading laboratory experiment.

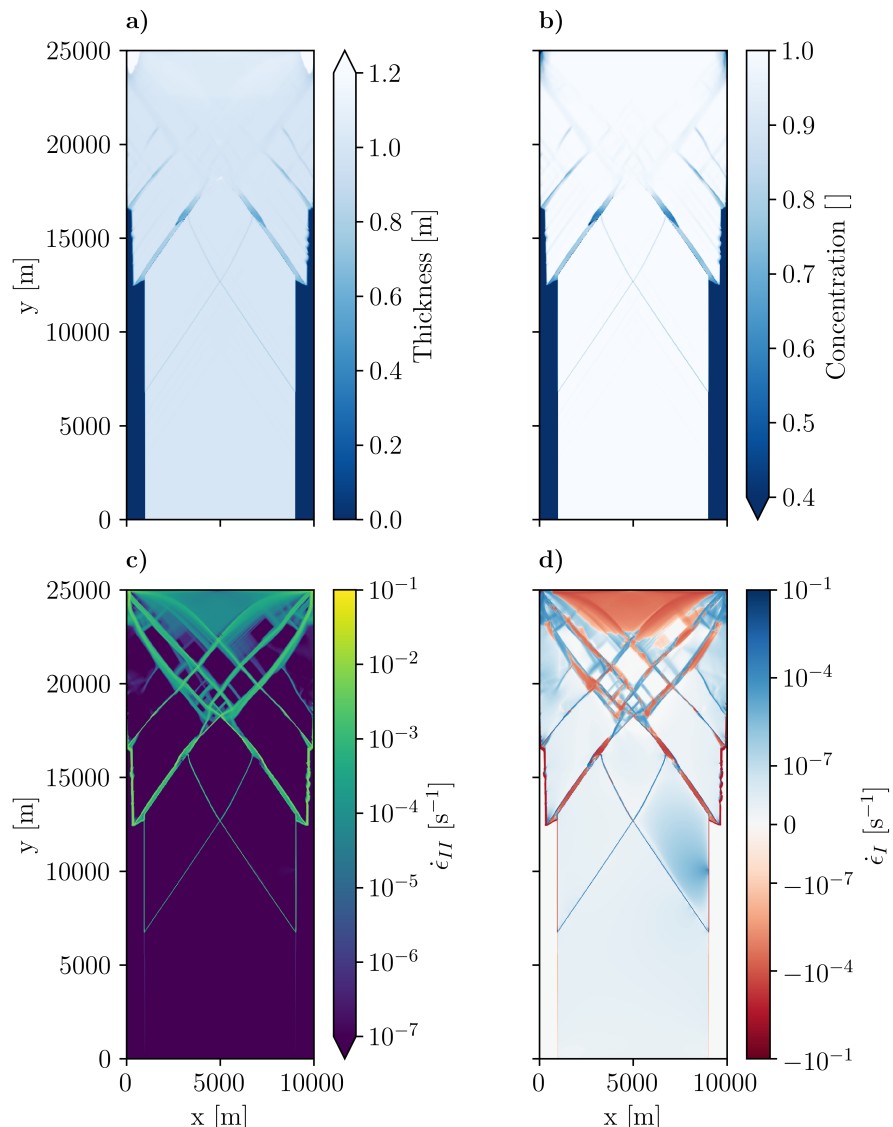

**Figure 6.** Sea ice thickness (a), concentration (b), maximum shear strain rate (c) and divergence (d) after 45 min of integration (2700 sec) in a uni-axial loading test. o make these longer simulations possible, both non-linear and linear iterations are limited to 150 per timestep. Results show the development of secondary fracture lines in all fields after the first fracture line has formed.

.

### 3.2.3 Effects of the heterogeneity

So far, all initial conditions have been homogeneous in thickness and concentration within the ice floe. In practice, sea ice (in a numerical model, but also in reality) is not homogeneous. A local weakness in the initial ice field is likely the starting point of

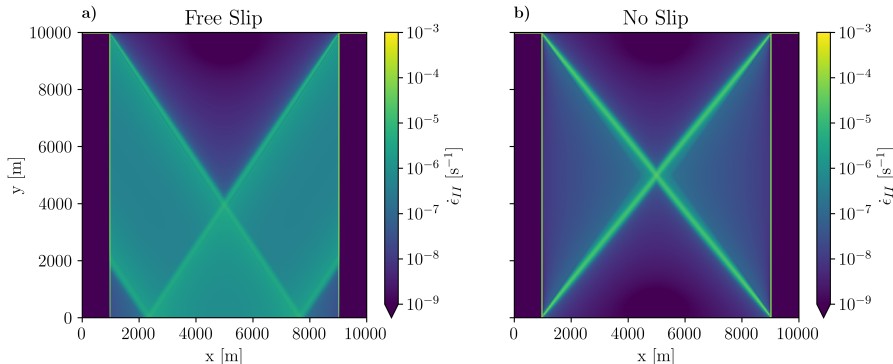

**Figure 7.** Maximum shear strain rate after 5 seconds of integration in a reduced size domain (8 km × 10 km) with free-slip (a) and no-slip (b) boundary conditions. Note that the no-slip boundary condition forces the fracture to occur at the corner of the domain, leading to a larger angle of $\theta = 39°$ vs. $34 \pm 1°$ in the control experiment. This suggests that the choice of boundary conditions in current sea ice model needs to be revisited.

a crack within the ice field (e.g., Herman, 2016, her Figure 5c). Local failures raise the stress level in adjacent grid cells and a crack can propagate. Note that the crack propagation in an "ideal" plastic model such as the VP model is instantaneous and this propagation is not seen between time steps. As a consequence, lines of failure will likely develop between local weaknesses. The location of weaknesses in the ice field together with the ice rheology (yield curve and flow rule) both determine the fracture

angles (Hibler and Schulson, 2000; Aksenov and Hibler, 2001). The influence of previous leads on subsequent lead creation have been studied with a Discrete-Element-Model (Wilchinsky et al., 2011) and has been used to constrain new anisotropic rheologies that include the effects of embedded anisotropic leads (Wilchinsky and Feltham, 2011, 2012).

To illustrate this behavior, we start new simulations from an initial ice field with two areas of zero ice thickness and zero ice concentration, hence weaker ice (Figure 9a). After $5\,\text{s}$ these simulations yield fracture patterns that are dramatically different

from those of the control run simulation (Section 3.1): the fracture lines now start and terminate at the locations of the weak ice areas. Still, changing the shear strength of the ice (by changing $e$) changes the fracture pattern (Figure 9b and c). With $e = 1$, the angles are much wider than with $e = 2$, which is consistent with the general dependence of fracture angles on $e$ (see Sec. 3.3.1). Our simulations cannot lead to conclusive statements about the relative importance of heterogeneity of initial conditions and yield curve parameters for the fracture pattern, but we can state that both affect the simulations in a way that

requires treating them separately to avoid confounding effects. Details are deferred to a dedicated study.

## 3.3   Effects of the yield curve on the fracture angle

### 3.3.1   Elliptical yield curve

Keeping $P^\star = 27.5\,\text{kN}\,\text{m}^{-1}$ at its default value, the maximal shear strength $S^\star = \frac{P^\star}{2e}$ is varied by changing the ellipse ratio $e$. Scaling the absolute values of $P^\star$ and $S^\star$ while keeping $e$ constant does not change the fracturing pattern as the tangent to

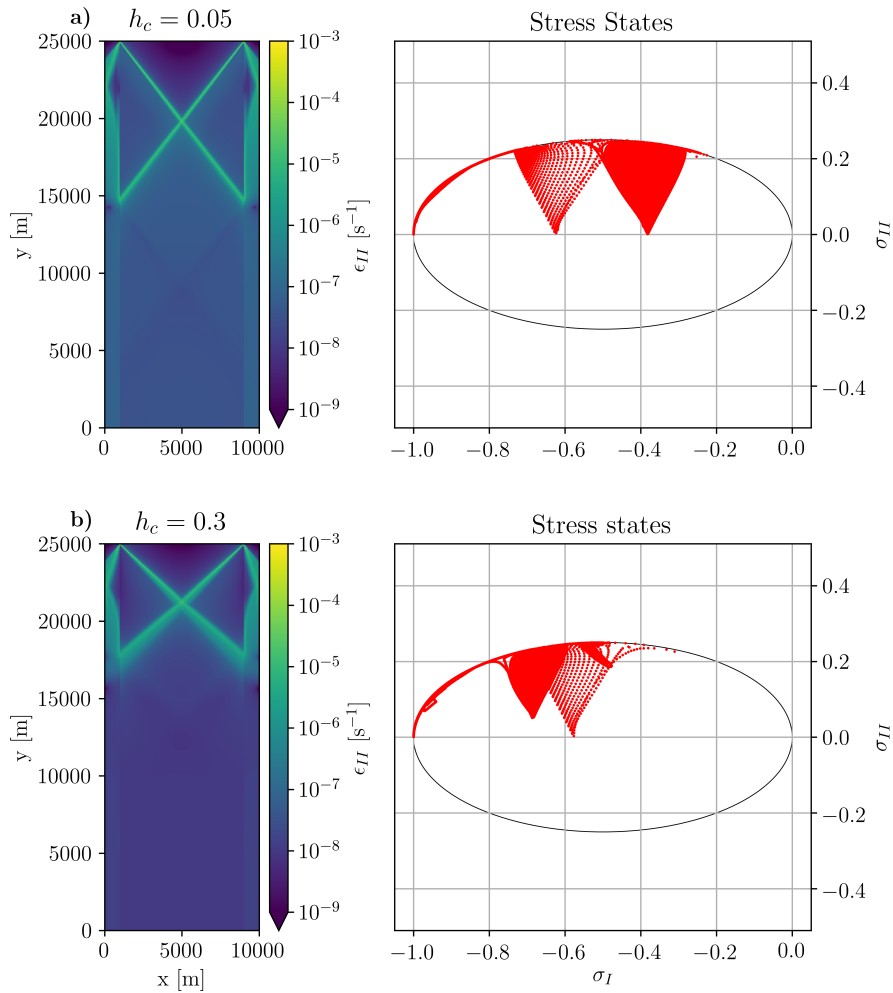

**Figure 8.** Maximum shear strain rates (left) and stress state in stress invariant space (right) after 5 seconds of integration for different confinement pressure: $h_c = 0.05\,\mathrm{m}$ (a) and $h_c = 0.3\,\mathrm{m}$ (b). Note, how stress states with divergent strain rates (a) migrate left towards convergent strain rates (b).

the ellipse stays the same (not shown). Changing the ellipse aspect ratio $e$ has a large effect on the fracture angle. The fracture angle decreases monotonically as the shear strength of the material (or $e$) decreases, from $61°$ for $e = 0.7$ to $32°$ for $e = 2.6$. This is clearly inconsistent with the behaviour of a granular material; in the sand castle analogue this would correspond to a dry sand castle with steeper walls than a moist sand castle. From the simple schematic of Fig. 4 it becomes clear that with increasing $e$ the intersection of the $\sigma_2$ axis with the yield curve gradually migrates from the left side of the ellipse to the right where the normal to the yield curve points increasingly towards convergent motion. We present a theoretical explanation for the sensitivity of the fracture angle to the shear strength of the material ($e$, for the ellipse) in Appendix B by re-writing the elliptical yield curve in local coordinates in the fracture plane ($\sigma$, $\tau$) instead of principal or stress invariant coordinates. The

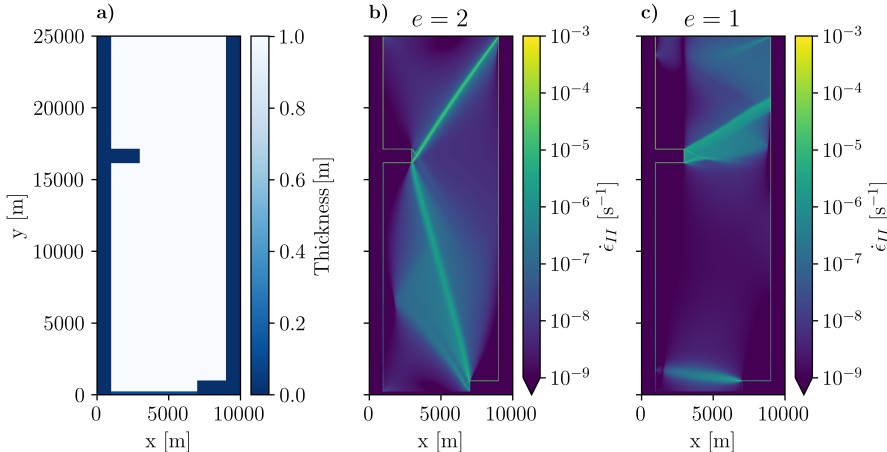

**Figure 9.** Sea ice thickness with two ice-free areas (a), and maximum shear strain rates for two different ellipse aspect ratios (b and c) after 5 seconds of integration. The position of the ice weaknesses determines the location and angle of the fracture lines, and also the rheology parameter $e$ has an entirely different effect. The main fractures lines are at angles of $25°$ and $34°$ for $e = 2.0$, and $57.6°$ for $e = 1.0$.

fracture angle is then determined from the slope of the tangent to the yield curve in local coordinates and this angle follows from the Mohr's circle (see for instance Popov, 1976).

Bouchat and Tremblay (2017) suggest a smaller ellipse aspect ratio (e.g. $e = 0.7$) to obtain a closer match with RADARSAT-derived distribution of deformation rates in pan-Arctic simulations at 10 km resolution. From Fig. 10 and 11, the corresponding

fracture angle is $\theta = (61\pm1)°$, that is, much larger than that is derived from RADARSAT images. $e$ also changes the distribution of the stress states on the yield curve. As the stress state migrates along the principal stress $\sigma_2$ until it reaches the yield curve in our uni-axial compressive test, the stress state are in the second half of the ellipse for $e < 1$ and the resulting deformation is in convergence (or ridging). The ice thickness increases due to ridging along the shear lines (Figure 11). In a longer simulation with $e = 0.7$ (not shown) the ice does not open but only ridges, with thicker ice building up within the ice floe. This is in strong

contrast to the results with $e = 2.0$ presented in Sect. 3.2.1, where the initial floe breaks up and separate floes form.

### 3.3.2   Coulombic yield curve

In this section, we replace the elliptical yield curve with a Coulombic yield curve (Hibler and Schulson, 2000). This yield curve consists of a Mohr-Coulomb failure envelope — two straight limbs in principal or stress invariant space with a slope $\mu$ — capped by an elliptical yield curve for high compressive stresses. Note that the flow rule applies only to the elliptical cap

in this yield curve. For the two straight limbs, the yield curve is normal to the truncated ellipse with the first stress invariant $\sigma_I$. For a Mohr-Coulomb yield curve, the fracture angle depends directly on the slope of the Mohr-Coulomb limb of the yield curve. Appendix A provides a theoretical explanation of how the angle of fracture depends on the internal angle of friction.

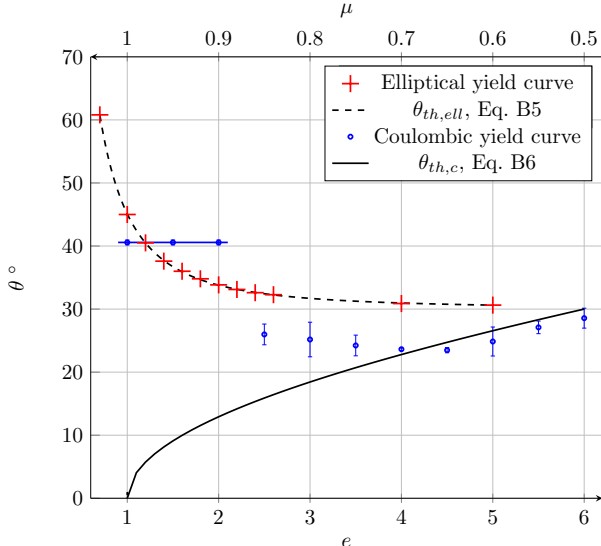

**Figure 10.** Fracture angles as a function of ellipse aspect ratio $e$ with constant P$^\star$ (red, bottom scale, Section 3.3.1). The theoretical relationship $\theta_{th,ell} = \frac{1}{2}\arccos{-1}\left[\frac{1}{2}\left(\frac{1}{e^2}-1\right)\right]$ (dashed black curve, Eq. B4 in the appendix) fits the modeled angles almost perfectly with $R^2 = 0.9995$ and $\sqrt{\text{VAR}} = 0.089$. The simulated fracture angles for the Coulombic yield curve as a function of the slope of the Mohr-Coulomb limbs (blue, top scale, Section 3.3.2) fit the theoretical relationship $\theta_{th,c} = \frac{1}{2}\arccos{-1}(\mu)$ only for $\mu \leq 0.7$ (black line, Eq. B5 in the appendix). The errors bars mean that they were more than one unique fracture line: For a small $\mu$, the ice breaks easily along the lateral edges of the floe. For $\mu > 0.7$ ($\phi = 44°$), the ambiguity appears because the stress states are both on the linear limbs and on the elliptical cap. For $\mu \geq 0.9$ (blue line), the fracture angle is the same as for the ellipse for $e = 1.4$.

The slope of the Mohr-Coulomb limbs of the Coulombic yield curve $\mu$ is varied between 0.3 and 1.0 (corresponding to an internal angle of friction $\phi = \arcsin(\mu)$ of 17.5° to 90°) to study how the fracture angle depends on the shear strength of the material. In all experiments with the Coulombic yield curve, we use a tensile strength of 5% of P$^*$ and an ellipse ratio $e = 1.4$, following Hibler and Schulson (2000). The tensile strength is introduced mainly for numerical reasons. With zero

5  tensile strength, the state of stress in a simple uni-axial compressive test with no confinement pressure is tangential to the yield curve at the origin (failure in tension) and on the two straight limbs (failure in shear) simultaneously, resulting in a numerical instability. With tensile stress (or confinement pressure) included, the state of stress reaches the yield only on the two limbs of the yield curve (see Fig. 12a).

For the Coulombic yield curve, there are two distinct regimes of failure. When the $\sigma_2$ axis intersects the yield curve on the

10  two straight limbs, which happens for our configuration for angles of friction $\phi < 45°$ (Fig. 12a, left hand side for $\mu = 0.7$ or $\phi = 44°$), the angle of fracture $\theta = \pi/4 - \phi/2$ as per standard theory (Appendix A). When the $\sigma_2$ axis intersects the yield curve on the elliptical cap, which happens for $\phi > 45°$ (Fig. 12c, for $\mu = 0.95$ or $\phi = 72°$), we observed a discontinuity in the fracture angle associated with the non-differentiable corner in the yield curve. Note that this corner cannot be removed (by

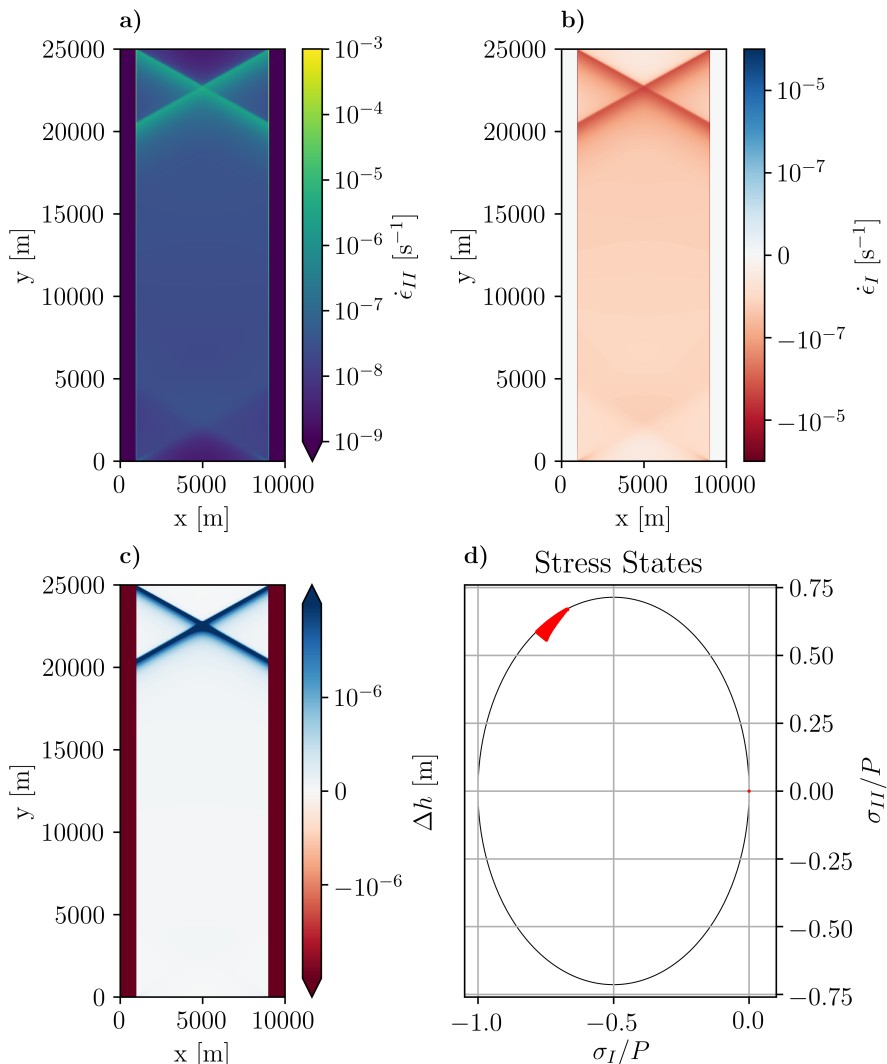

**Figure 11.** Maximum shear strain (a), ice thickness anomaly (b), divergence (c) and stress state in stress invariant space (d) after 5 sec of integration for a smaller ellipse aspect ration ($e = 0.7$ compared to $e = 2$ in the reference run in Sect. 3.1). Compared to the control run on Fig. 3, the angle of fracture is larger ($\theta = (61 \pm 1)°$), the stress states are in the second half of the ellipse (with strain rates pointing into the convergent direction) and there is convergence along the fracture lines (panel b) in agreement with the schematic in Fig. 4.

changing the $P^\star$ and $e$ of the elliptical cap) as the two straight Mohr-Coulomb limbs are defined as a truncation of the ellipse. For $\phi \approx 45°$ in our configuration, the numerical solver has difficulties reaching convergence because of the non-differentiable corner in the yield curve between the elliptical cap and the two straight limbs (Fig. 12b, middle panel for $\mu = 0.8$ or $\phi = 53°$). Finally for very small angles $\phi$, a large number of fractures, as opposed to single well defined fracture lines, appears because of the weakness of the material in shear. This behaviour is not something that is typically observed in uni-axial compressive

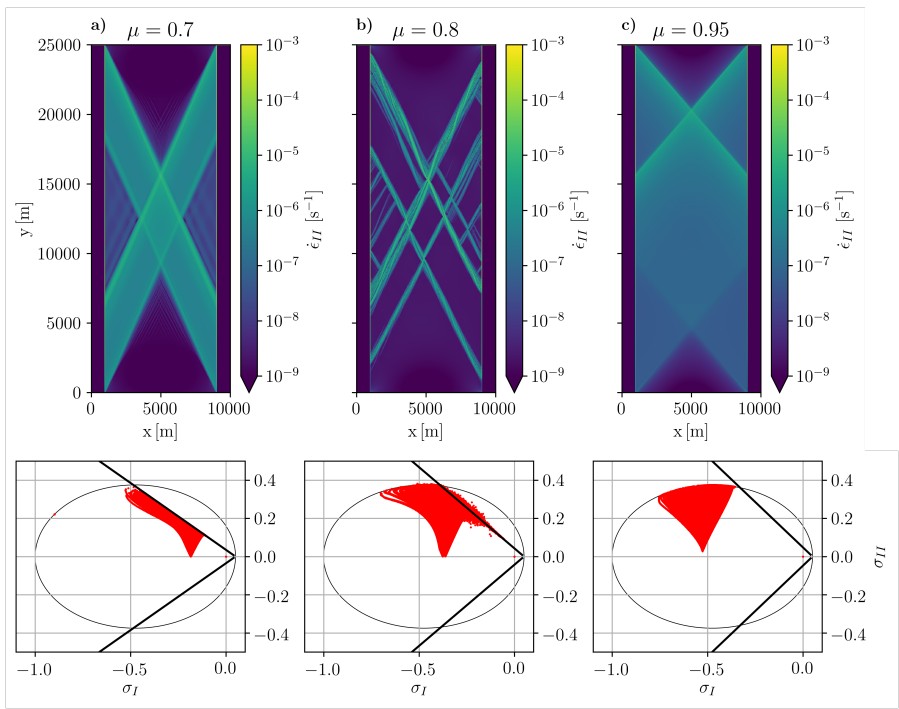

**Figure 12.** Maximum shear strain (top) and stress state in stress invariant space (bottom) for different internal angles of friction. (a) $\mu = 0.7$ or $\phi = 44°$, (b) $\mu = 0.85$ or $\phi = 58°$ and (c) $\mu = 0.95$ or $\phi = 72°$ after 5 s of integration. The angles of fracture are $\theta = 23°$, $(28 \pm 2)°$ and $41°$. Fig. 10 illustrates how $\theta$ depends on $\mu$ for a Coulombic yield curve.

test of a granular material who generally have higher shear resistance. Note that the value of $\phi$ that are characteristic for the individual regimes depends on the amount of tensile strength.

## 4 Discussion

Our idealized experiments using the VP rheologies resolve fracture lines as described by Hutchings et al. (2005) and akin to observations (Kwok, 2001). The fracturing of the ice floe creates smaller floes in a manner that appears realistic, for example, compared to to Landsat-7 images (Schulson, 2004, Figure 2). At the high resolution of 25 m the original interpretation of the continuity assumption, namely that each grid cell should represent a distribution of floes (Coon et al., 1974), is no longer valid, but we show that the fracture angle is independent of resolution and scale as expected. Instead, the emerging discontinuities and the polygonal diamond shape of the fracture lines that appear as floes spanning many grid cells are a consequence of the mathematical characteristics of the VP model (Pritchard, 1988). Diamond shaped floes are observed in the Arctic ocean (Erlingsson, 1988; Walter and Overland, 1993) and also modeled using a Discrete Element Model (DEM) in an idealized experiment (Wilchinsky et al., 2010). The Elastic Anisotropic Plastic (EAP) rheology assumes predominately diamond shaped

floes in sea ice (Wilchinsky and Feltham, 2006). A sea ice model with EAP creates sharper fractures than a model with the Elastic Viscous Plastic (EVP, Hunke and Dukowicz, 1997) rheology (Heorton et al., 2018). The authors concluded that the anisotropic model may improve the fracturing process for sea ice, especially by creating areas of oriented weaknesses, and particularly at coarse resolution where the fracture is not resolved by the grid spacing. In the experiments presented here, the VP rheologies lead to sharp and anisotropic fracture lines without any additional assumptions.

We explored some experimental choices to separate their effects from those of the rheology parameters. The fracture angles do not depend on the spatial resolution and domain size as expected in our idealized numerical experiment setup (Sect. 3.2.1, Fig. 5). The maximum viscosities in the VP model are very high and consequently, the VP model can be considered as an ideal plastic material (i.e. a model with an elastic component that has an infinite elastic wave speed). For this reason, fracture in a VP model occurs almost instantaneously. Observed time scales of fracture are on the order of 10 seconds for 60 m floe diameters (Dempsey et al., 2012, Figure 6 top right panel) and from typical elastic wave speeds of 200–2000 $\mathrm{m\,s^{-1}}$, large cracks of order 1000 km can form in minutes to hours (Marsan et al., 2012).

In our setup, the no-slip boundary condition has little effect on the fracture pattern, but our results suggest that in basin-wide simulations the choice of boundary conditions affects the fracture depending on the geometry and stress direction. The no-slip condition appears to be unphysical. It acts to concentrate the stress on the corners of the floe and forces the fracture to occur at this location. This should motivate a more thorough investigation of the boundary conditions for LKFs that form between one shoreline and another. Similar results were obtained from analytical solutions in idealized geometry for the Morh-Coulomb yield curve with a double sliding deformation law (Sirven and Tremblay, 2014).

The confining pressure (i.e. thin ice imposed on the side of the domain) changes the distribution of stress within the domain. This results in different deformation patterns (shear and divergence) and different fracture angles because the yield curve is convex and uses a normal flow rule. From this we can conclude that by surrounding our floe with open-water, we get the most acute angles from the rheology in this uni-axial compression setup. This is not consistent with the behavior of typical granular material for which an angle of fracture is independent of the confining pressure (Hutter and Rajagopal, 1994). Details of a heterogeneous ice cover also affect the fracture pattern. LKFs link the weaknesses in the ice cover, but the pattern still depends on the preferred fracture angles implied by the model rheology. In summary, we are confident that our choice of parameters allows us to isolate the effects of the rheology and the yield curve on the fracturing process.

In granular material, large shear resistance is linked to contact normals between floes that oppose the shear motion and lead to dilatation (Balendran and Nemat-Nasser, 1993). In our experiments, increasing shear strength in the standard VP model (reducing the ellipse aspect ratio $e$) does not decrease, but increases the fracture angle. This is in contrast to the behavior of granular material where larger shear strength leads to lower fracture angles — think of a moist sand castle versus a dry sand castle. In addition, high shear strength in the VP model with the elliptical yield curve leads to convergence along the fracture plane whereas observations (e.g. RADARSAT derived deformation fields) show a range of positive and negative divergence along LKFs — in accordance with laboratory tests of granular material that show a variable internal angle of friction that depends on the distribution of the contact normals between individual floes (Hutter and Rajagopal, 1994). Inspection of the stress states in the 2D stress plane suggests that the intersection of the yield curve with the $\sigma_2$ axis has an important role in

the fracture process. This intersection point appears to determine the fracture angle. In fact, the angle is determined from the intersection of the Mohr's circle of stress with the yield curve to give a theoretical relationship between the fracture angle and the ellipse ratio $e$. With our experiments, we were able to confirm this relationship empirically.

Arctic-wide simulations improve metrics of sea ice concentration, thickness and velocity by decreasing the value of $e$ of the standard elliptical yield curve, that is, by adding shear and bi-axial tensile and compressive strength (Miller et al., 2005; Ungermann et al., 2017). The representation of sea ice arches improves with smaller $e$ (Dumont et al., 2009) so do LKF statistics (Bouchat and Tremblay, 2017). Our results, however, show that this makes the fracture angles larger, which is in stark contrast to what we expect to be necessary to improve the creation of LKFs in sea ice models.

The fracture angle and the sea ice opening and ridging depending on the deformation states are consistent with the theory of the yield curve analysis developed in Pritchard (1988) and the Mohr's circle framework that we present in Appendix B. Interestingly, a change of ice maximum compressive strength $P^\star$ with a constant $e$ has no influence on the LKF creation, although $P^\star$ is usually thought of as the principal parameter of sea ice models in climate simulations (e.g. Schmidt et al., 2014). The effects of bi-axial tensile strength $T^\star$ on fracture processes require further investigation, especially given the fact that the assumption of zero-tensile strength is being challenged (Coon et al., 2007). The ice strength parameter $C^\star$ (the parameter governing the change of ice strength depending on ice concentration, Equation 4) was not studied here, although it appears to be an important tuning parameter and it also helps to improve basin-wide simulations (Ungermann et al., 2017). The simulations presented in this study are not realistic and cannot be compared directly to observations of ice floe fracture. For instance, our idealized ice floe is homogeneous while sea ice is known to feature some weaknesses like thermal cracks or melt ponds.

With the Coulombic yield curve, the simulated fracture angle can be smaller than for the elliptical yield curve. For $\mu = 0.7$ ($\phi = 44°$) theory predicts $\theta_{MC} = 22.8°$ (Appendix B). The simulated fracture angle with $\mu = 0.7$ of $\theta = 23.5°$ is close to the $\simeq 20°$ described in Hibler and Schulson (2000). Erlingsson (1988) developed a different Mohr-Coulomb theory linking internal angle of friction and fracture angle. This complex theory takes into account the fractal (or self-similar) nature of sea ice. It gives different results, but is inadequate for a single ice floe simulated as presented here. Based on the results of Pritchard (1988), Wang (2007) used observed fracture patterns to design a *Curved Diamond* yield curve. But this yield curve also contains a non-differentiable point, which will be problematic for numerical reasons. The Coulombic yield curve used here uses a normal flow, and consequently divergence will always be present along shear lines. In-situ measurements, however, show that the deformations follow a non-normal flow rule (Weiss et al., 2007) and large-scale observations show both divergence and convergence (ridges) along LKFs (Stern et al., 1995). There are alternative flow rules still to be explored, for example a double-sliding law with (Ip et al., 1991) or without dilatation included (Balendran and Nemat-Nasser, 1993; Tremblay and Mysak, 1997).

## 5  Conclusions

Motivated by the observation that the intersection angles in a 2 km Arctic-wide simulation of sea ice are generally larger than in the RGPS dataset (Hutter et al., 2018b), the fracturing of ice under compression was studied with two VP rheologies in a highly idealized geometry and with very small grid spacing of 25 m. The main conclusions are:

In our experimental configuration with uni-axial compression, fracture angles below $30°$ are not possible in a VP-model with an elliptical yield curve. Observations suggest much lower values. We find an empirical relationship between the fracture angle and the ellipse ratio $e$ of the elliptical yield curve that can be fully explained by the convexity of the yield curve (Appendix B). In contrast to expectations, increasing the maximum shear strength in the sea ice model increases the fracture angle. Along a fracture line, there can be both divergence and convergence depending on the shear strength of the ice, linked to the flow rule.

The simulated ice opens and creates leads with an ellipse ratio $e > 1$ (shear strength is smaller than compressive strength), and ridges for $e < 1$ (shear strength is larger than compressive strength).

With a modified Coulombic yield curve, the fracture angle can be decreased to values expected from observations, but the non-differentiable corner points of this yield curve lead to numerical (convergence) issues and, for some values of the coefficient of internal friction $\mu$, to fracture patterns that are difficult to interpret. At these corner points, two different slopes meet and

give two non-unique solutions for fracture angles and deformation directions. We recommend to avoid non-differentiable yield curves (with a normal flow rule) in viscous-plastic sea ice models.

More generally, the model produces diamond-shaped fracture patterns. Later the ice floe disintegrates and several smaller floes develop. The fracturing process in the ice floe in our configuration is independent of the experiment resolution and scale, but sensitive to boundary conditions (no-slip or free-slip). The fracture angle in the VP-model is also sensitive to the confining

pressure. This is not consistent with the notion of sea ice as a granular material. Unsurprisingly, the yield curve plays an important role in fracturing sea ice in a numerical model as it governs the deformation of the ice as a function of the applied stress.

The idealized experiment of a uni-dimensional compression is useful to explore the effects of the yield curve because all other parameters are controlled. Historically, the discrimination between the different yield curves was not possible because of the

scarcity of sea ice drift data. Model comparisons to recent sea ice deformation datasets, such as from RADARSAT, imply that we would need to increase the shear strength with the ellipse in the standard VP rheology to match observations (Bouchat and Tremblay, 2017). We find that this increases the fracture angles, creating a dilemma. Therefore, the high-resolution idealized experiment presented in this work provides a framework to investigate and discriminate different rheologies — yield curve and flow rule.

If Arctic-wide sea ice simulations with a resolution of 25 m are not feasible today because of computational cost, we can still imagine small experiments to be useful for process modeling on small scales when local and high-resolution observations (e.g. wind, ice velocities) are available. For example, such process modeling studies could be used to constrain the rheology with data from the upcoming MOSAiC campaign (Dethloff et al., 2016) that will provide a full year of sea ice observations in pack ice. Such simulations would also need to take into account the effects of heterogeneous ice cover and wind patterns, with

potentially convergent and divergent wind forcing. Most climate models use the standard VP rheology (Stroeve et al., 2014) or one of its variants (e.g. EVP). Results presented here, however, imply that a more physical yield curve with a (possibly non-associative) flow rule is required. Such a yield curve would have to be continuous in all representations, differentiable without corners, have some cohesion and be consistent with available observations of fracture angles in convergent and divergent flow.

## 5 Appendix A: Fracture angle

Below, we derive a relationship between the fracture angle and the internal angle of friction for a Mohr-Coulomb yield criterion for completeness. We consider an arbitrary piece of a 2D medium (Figure A1a) that is subject to stresses in physical stress space $\sigma_{ij}$ $(i = 1, 2)$ . Computing the change of coordinates as described in Eq. (7), we can consider the principal stresses $(\sigma_1, \sigma_2)$ applied on the medium (Figure A1b). From the force balance, the normal stress $\sigma$ and the shear stress $\tau$ on a plane at an angle
$\theta$ from the principal stress axis can be written as (see Fig. A1b and Popov, 1976)

$$\sigma dA = \sigma_2 \sin(\theta) \sin(\theta) dA + \sigma_1 \cos(\theta) \cos(\theta) dA, \tag{A1}$$

$$\tau dA = -\sigma_2 \cos(\theta) \sin(\theta) dA + \sigma_1 \cos(\theta) \sin(\theta) dA, \tag{A2}$$

where $dA$ is the area of the friction plane on which the stresses are applied (in 2D it is just a line). The second trigonometric term comes from the fact that this surface is tilted compared to the direction of stresses $\sigma_1$ and $\sigma_2$. Using angle sum and
15 difference identities of trigonometry, we can write the stresses $\sigma$ and $\tau$ in terms of the principal stresses $\sigma_1$ and $\sigma_2$ as

$$\sigma = \frac{1}{2}(\sigma_1 + \sigma_2) + \frac{1}{2}(\sigma_1 - \sigma_2)\cos(2\theta), \tag{A3}$$

$$\tau = \frac{1}{2}(\sigma_1 - \sigma_2)\sin(2\theta). \tag{A4}$$

In terms of the stress invariants $\sigma_I$ and $\sigma_{II}$ this gives

$$\sigma = \sigma_I + \sigma_{II}\cos(2\theta), \tag{A5}$$

$$\tau = \sigma_{II}\sin(2\theta). \tag{A6}$$

The Mohr-Coulomb failure criterion can be written in the fracture plane stress space (see Fig. A2) as

$$\tau = -\tan(\phi)\sigma + c, \tag{A7}$$

where $\phi$ is the internal angle of friction, and $c$ the cohesion when no stresses are applied (Verruijt, 2018).

Substituting (A5) and (A6) in (A7) we get

$$\sigma_{II}\sin(2\theta) = -\tan(\phi)\sigma_I - \tan(\phi)\sigma_{II}\cos(2\theta) + c, \tag{A8}$$

and after multiplying both sides by $\cos(\phi)$

$$\sigma_{II}\left[\sin(2\theta)\cos(\phi) + \cos(2\theta)\sin(\phi)\right] = -\sigma_I\sin(\phi) + c\cos(\phi). \tag{A9}$$

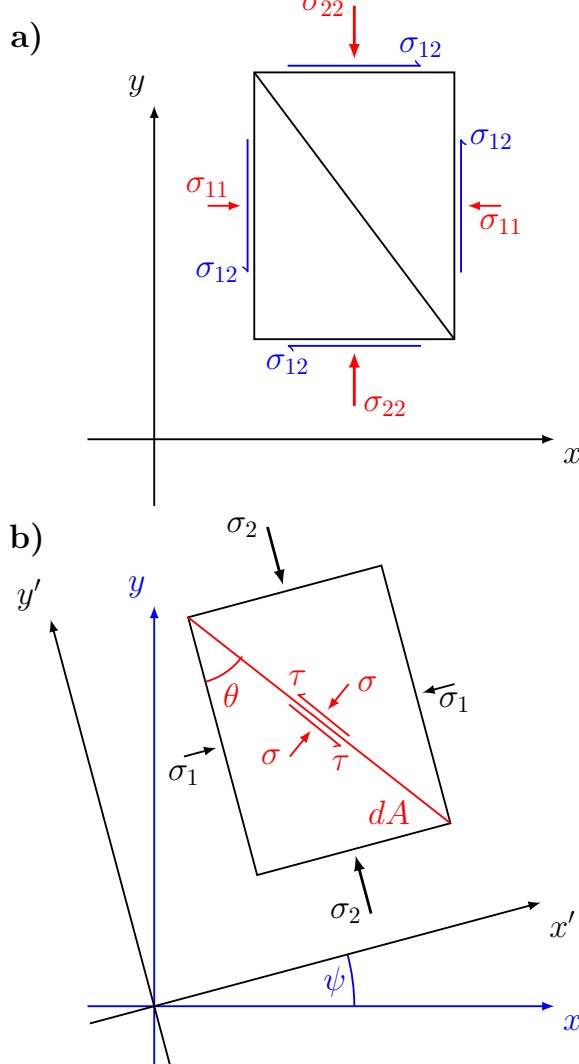

**Figure A1.** Stress state in physical stress space (a) and in an arbitrary coordinate system oriented at an angle *theta* with respect to the principal stress axes (b). The principal stresses are the eigenvalues of the stress tensor in an arbitrary coordinate system and the angle $\psi$ is derived from the rotation matrix composed of the two eigenvectors. Note that in the study above there is no shear stress ($\sigma 12 = 0$, so principal axes and physical axes are aligned ($\psi = 0$).

By geometrical construction (see Fig. A2) the MC criterion is satified when (see also Verruijt, 2018, Sect. 20.4),

$$\sigma_{II} = -\sigma_I \sin(\phi) + c \cos(\phi) \tag{A10}$$

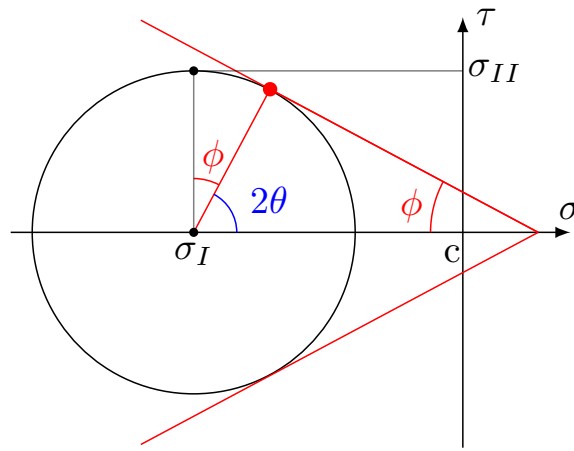

**Figure A2.** Mohr's circle of stress (black) with Mohr-Coulomb yield criterion (red) of angle of internal friction $\phi$ (red) and cohesion c in $(\sigma,\tau)$ space. From Eq. (A12, the deformation is created with an angle $\theta$ that can be represented in Mohr's circle (blue).

so that eq. (A9) becomes

$$\sin(2\theta)\cos(\phi) + \cos(2\theta)\sin(\phi) = \sin(2\theta + \phi) = 1, \tag{A11}$$

from which we get the classical result of material deformation physics:

$$2\theta + \phi = \frac{\pi}{2} \Rightarrow \theta = \frac{\pi}{4} - \frac{\phi}{2} \tag{A12}$$

## Appendix B: Fracture angle and yield curve

A yield curve can be defined in the local stress ($\sigma_{ij}$), principal stress ($\sigma_{1,2}$) or stress invariant ($\sigma_{I,II}$) spaces. The latter gives the center and radius of the Mohr's circle of stress defining all equivalent stress states ($\sigma,\tau$) for all angles with respect to a reference coordinate system. This allows the translation of the elliptical yield curve from the standard principal or stress invariant space to a local stress coordinate system ($\sigma_{ij}$). In this sense, we can plot the yield curve in ($\sigma,\tau$) space as the envelope of all Mohr's circles for each point on the elliptical yield curve defined in stress invariant coordinates (see Fig. B1 for an illustration with the elliptical yield curve). In the following, we refer to this envelope of all Mohr's circles as the reconstructed yield curve. The tangent to this curve can be expressed as (Figure B2):

$$\sin(\phi) = \tan(\gamma) = \mu = \frac{\partial \sigma_{II}}{\partial \sigma_I}. \tag{B1}$$

We can then express the fracture angle for stress states on the yield curve envelope by placing Eq. (B1) in Eq. (A12) :

$$\theta(\sigma_I) = \frac{\pi}{4} - \frac{1}{2}\arcsin\left(\frac{\partial \sigma_{II}}{\partial \sigma_I}(\sigma_I)\right) = \frac{1}{2}\arccos\left(\frac{\partial \sigma_{II}}{\partial \sigma_I}(\sigma_I)\right). \tag{B2}$$

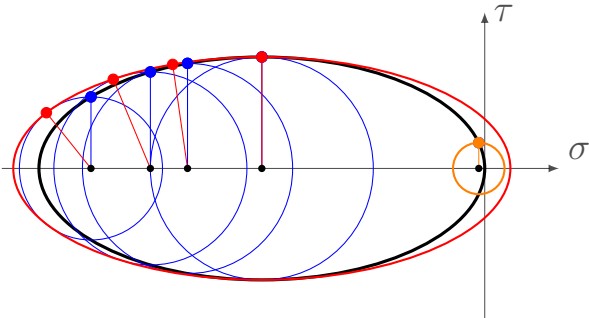

**Figure B1.** Illustration of the Mohr's circle applied to the elliptical yield curve (black ellipse) in $\sigma,\tau$ space, some examples of Mohr's circles (blue) and the reconstructed yield curve (red) in the fracture plane space. The orange Mohr's circle illustrate the case where no fracture lines exists, for $|\mu| > 1$.

This is the same relation presented (Pritchard, 1988) and used previously (Wang et al., 2006), but obtained within the $(\sigma,\tau)$ stress space.

## B1   Elliptical yield curve

From the previous equations, some implications about the elliptical yield curve immediately follow. As shown in Fig. 4, in a

uni-directional compressive setup the slope of a tangent to the yield curve changes with the ellipse ratio. The convexity of the ellipse implies that the ratio $\frac{\tau}{\sigma} = \tan(\phi)$ of shear strength $\tau$ to compressive strength $\sigma$ becomes smaller with smaller $e$. If we compute the slope of the tangent to the elliptical yield curve at the intersection point between the yield curve and the $\sigma_2$ axis, we get

$$\left.\frac{\partial \sigma_{II}}{\partial \sigma_I}\right|_{\sigma_1=0} = \frac{1}{2}\left(\frac{1}{e^2} - 1\right). \tag{B3}$$

Inserted this relationship into Eq. (B2) gives the angle of fracture for uni-axial compressive experiment with an ellipse ratio $e$:

$$\theta_{th,ell}(e) = \frac{1}{2}\arccos\left[\frac{1}{2}\left(\frac{1}{e^2} - 1\right)\right]. \tag{B4}$$

Note that a yield curve in $(\sigma_{I,II})$ space with a tangent slope above unity does not have a Mohr's circle that can be tangent to the yield curve in $(\sigma,\tau)$ space (Orange circle on in Fig. B1). This implies that no angle of fracture can be derived for these stress states. This is the case for the elliptical yield curve for low and high compressive stresses. It is still unclear what happens

in the VP model for stress states on the yield curve that have a tangent with a slope higher than unity (see also Pritchard, 1988). Note also that for some $(\sigma_I,\sigma_{II})$ states, the ice will actually fail in tension, as the reconstructed yield curve having a few points in the first and fourth quadrant.

The shear and bulk viscosities are symmetrical about the center of the ellipse. This implies that they are equal for divergence and convergence. Clearly this is not physical since, for shear deformations where ice floes continue to interact with one another

(termed the quasi-static flow regime (Babić et al., 1990), divergent flow counter-intuitively should have more ice-ice interactions and higher viscosities than convergent flow — because divergent flow is the result of a higher number of contact normals opposing the shear. When the divergence is large and floes no longer interact, the shear and bulk viscosities are still symmetrical about the center of the ellipse. While this is non-physical, it does lead to more numerical stability because the extra viscosity or dissipation of energy regularizes the problem. We also note that a yield curve with a tangent that has a slope smaller than one (in absolute value) in the first and fourth quadrant (positive first principal stress) is unphysical because it would lead to a diamond shaped pair of ice fracture even in a uni-axial tensile test, which is inconsistent with laboratory experiments (Cox and Richter-Menge, 1985; Menge and Jones, 1993). We conclude that adding tensile strength to the elliptical yield curve may not be physical. The behavior of the elliptical yield curve in uni-axial tensile tests will be explored elsewhere.

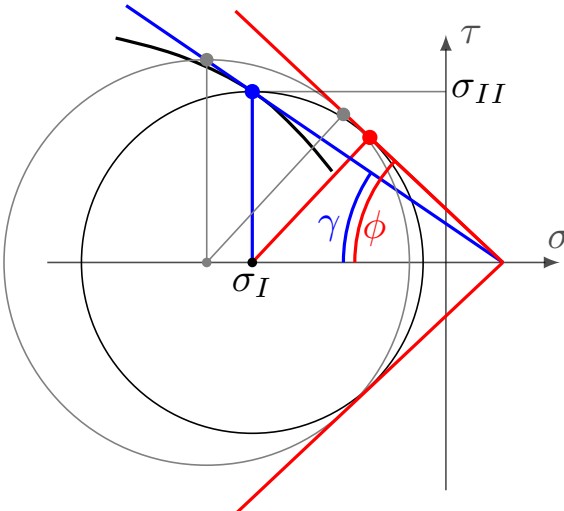

**Figure B2.** Mohr's circle of stress with an arbitrary yield curve (black line) in the fracture plane reference. $\tan(\gamma) = \mu$ is the tangent to the yield curve, and $\phi$ the internal angle of friction as described in Appendix A. We note that $\sin(\phi) = \tan(\gamma) = \mu$ ( for $|\mu| \leq 1$). For a slightly different Mohr's circle (grey), the blue and and red tangents meet in the same point on the $\sigma$ axis.

## B2 Coulombic yield curve

Applying Mohr's circle to the Coulombic yield curve explains, why the non-differentiable corners in the yield curve lead to numerical problems (Figure B3). The tangent does not vary smoothly and the reconstructed yield curve in the failure plane $(\sigma, \tau)$ becomes discontinuous (Figure B3, red line). As shown in Sect. 3.3.2, when the stress states fall on only one of the two parts (ellipse or limb) the conjugate faults forms as expected. Using Eq. (B4) with $\mu$ as the slope of the Mohr-Coulomb limbs of the Coulombic yield curve, the fracture angle is given by

$$\theta_{th,c}(\mu) = \frac{1}{2}\arccos(\mu), \tag{B5}$$

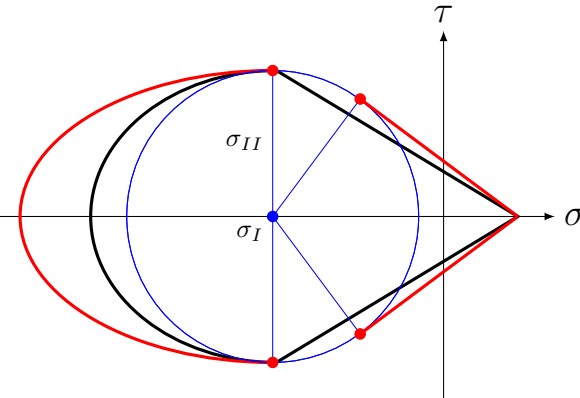

**Figure B3.** Mohr's circle applied to the coulombic yield curve (in black) in $\sigma,\tau$ space, the Mohr's circle for the cusps between the elliptical cap and the Mohr-Coulomb linear limbs (blue circle), and the yield curve in $(\sigma,\tau)$ space (red). We can see the effect of combining two regimes, for the same Mohr's circle, two different angles coexists (red circles) and are apart from each other.

which is identical to Eq. (A12).

   The Coulombic yield curve with an internal angle of friction of one ($\mu = 1$) and no cohesion ($c = 0$) (also called the Truncated Ellipse Method, TEM Hibler and Schulson, 1997, Appendix) only has one possible solution with an angle of fracture equal to 0 degree (i.e. conjugate pair of fracture are not possible). Zero cohesion implies that the ice will deform even for nearly
no stress. This yield curve also appears unphysical to us.

*Author contributions.* Damien Ringeisen designed the experiments, ran the simulations, interpreted the results with the help from Martin Losch and Bruno Tremblay. Nils Hutter contributed to the discussion on LKFs in simulations and observations. Damien Ringeisen prepared the manuscript with contributions from all co-authors.

*Competing interests.* The authors declare that they have no conflict of interest.

*Acknowledgements.* We would like to thank Jennifer Hutchings and Harry Heorton for their helpful reviews on this paper, as well as Amélie Bouchat and Mathieu Plante for useful discussion during this work. This project was supported by the Deutsche Forschungsgemeinschaft (DFG) through the International Research Training Group "Processes and impacts of climate change in the North Atlantic Ocean and the Canadian Arctic" (IRTG 1904 ArcTrain). The authors would like to thank the Isaac Newton Institute for Mathematical Sciences for support and hospitality during the program Mathematics of sea ice phenomena when work on this paper was undertaken. This work was supported
by: EPSRC grant numbers EP/K032208/1 and EP/R014604/1. This work is a contribution to the Canadian Sea Ice and Snow Evolution

(CanSISE) Network funded by the Natural Sciences and Engineering Research Council (NSERC) of Canada, the Marine Environmental Observation Prediction and Response (MEOPAR) Network and the NSERC Discovery Grants Program awarded to Tremblay.

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
