# Peer review of "Simulating intersection angles between conjugate faults in sea ice with different VP rheologies"

_The Cryosphere, 2018_

## Referee Comment (RC1) · Heorton (Referee) · 11 Dec 2018

Modelling Sea Ice fracture at very high resolution with VP rheologies

This paper documents idealised very high resolution (25 m) numerical simulations of deformation in sea ice using the viscous plastic VP rheology. The sea ice is predominantly put under uniaxial compression from the top and bottom boundaries and the resulting deformation features are documented. Simulations are performed with longer model run times, decreased spatial resolution, modified boundary conditions, biaxial compression, imposed flaws within the sea ice and alternate rheologies by changing the yield curve shape. The results are well documented, though further detail is required on the model setup. Particular emphasis is put on the resulting deformation or

linear kinematic feature intersection angle, as a means to provide a link between simulated and observed deformation features, and thus provides insight into how to select an appropriate sea ice rheology for simulation within climate models or for future studies. This method presents an exciting way to link observations and simulations of sea ice deformation, but there are certain aspects of the study that, in my opinion, need addressing.

Firstly is selection of model resolution, and consideration of whether to simulate a single floe or a continuum of many floes. The selected 25 m resolution makes this a simulation of a single floe, and the paper describes the study as such, but there is little discussion of whether the selected VP rheology is a valid representation of a single floe. The VP rheology was developed to simulate a continuum of floes over ocean length scales, and it is not immediately obvious to me, that the VP rheology is a valid representation of the deformation of a single floe just by reducing the model resolution to that of single floe, and setting the ice concentration to unity as performed in this study. I am unaware of previous simulations of solid body deformation from other fields, and it may be the case that VP like rheologies are well studied, but this needs to be discussed in this paper. On the other hand this paper may be be a proof of concept that the VP rheology can be used to simulate the deformation of a single floe, and if it is, then it needs to be worded as such.

A further implication of simulating a single floe, rather than a continuum floes, is the observations that the paper discusses when validating the results. The RADARSAT and RGPS data is given as a means to validate the yield curve selected. But the RGPS data is given on continuum length scales, so to validate a selected yield curve the comparison will need to be made with simulations over continuum length scales, not on the floe length scales presented in this paper. To validate the simulations of this paper observations of individual floe shape, aspect ratio and floe-to-floe crack intersection angle from aerial images of floes will be best suited.

Secondly the presented results are often from the order of 1-10 seconds into the sim-

ulations. This is in contrast to idealised deformation experiments of Hutchings et al. (2004) and Heorton et al. (2018) where the results are given at 1 - 24 hours of simulation, and explored over multiple days. Within these studies the initial conditions and early stress/deformation states are explored and documented. Also in both of these cases, the initial state of the ice over was seeded with noise (in strength, or thickness and forcing) to allow for features to develop and stop an unrealistic uniform sea ice cover. Were such considerations performed in this study? Are the initial conditions uniform? What are the implications of using results from 1-10 seconds of model run? I would like to see documentation of the time series of stress/strain state in order to validate the idealised experiments and the initial conditions. I personally found the longer documented run of 45 minutes interesting, with individual floe-like shapes appearing that can be compared to images of floes.

Also there is little discussion of the implications and 'robustness' of the presented model results. The authors described the results as 'robust' on multiple occasions but make little effort to inform the reader why they are robust. Figure 5 is presented as a domain/resolution study but only mentioned once in passing as an indication of the models robustness. I'm assuming that the model domain/resolution/run time has been investigated but there is no documentation or discussion of the models limitations. A section describing this is required in order to allow the other results to be published.

Thirdly it is not obvious how the deformation or linear kinematic feature intersection angle were calculated. I would like to see this information given in an appendix, such a method is a very useful contribution to studies of sea-ice rheology and one that I would like to use in the future. A citation to another study where this method was performed and is described is another option. The given appendix showing the theory behind internal friction and Mohr-Coloumb failure criterion, whilst interesting to see, does not appear to be original theory for this paper and is not required and the dependencies can be stated and cited.

Particular comments: Page 1 Line 6-8 What are the dependencies of typical granular

materials? The sand castle analogy is not useful. Line 8, what model? this paper or previous? Line 10-14 More description of 'typical granular materials' that are not accurately described and all comparisons are difficult to follow. I would avoid these loose comparisons in the abstract and stick to definite results. Line 24 are the two citations model studies or observations of sea ice floes? Observational studies are required for this sentence.

Page 2 Line 5 Leads plural, dangerous to use the word memory when describing a computer model, 'emergent anisotorpy' is more accurate. Line 8 The equations are difficult to solve due to their non-linearity and complexity not because of sea ice. It is however difficult to represent sea ice with simple, easily solvable equations due to it's non-continuous features. 24 argues - argued 28 rheologies plural 32 check citations and parenthesis 33 check citations and parenthesis 34-35 'Based on these satellite observations, amongst others (provide some examples), and in-situ. . .'

Page 3 line 6 space before 'Girard' line 13 delete parenthesis line 15 'are appear as line of' - 'appear as lines of', deformation singular, 'with the deformation' - 'with shear deformation', divergence - convergence Line 18 leads - leading line 19-20 check citation parenthesis

Page 4 line 1 Wilchinsky et al. also deduced intersection angles between floes that are relevant to this paper.

Page 5 Figure 1. What is 'Mohr-Coloumb flat' In general I found the figure captions to be lacking in content. Can they all be expanded to directly describe what simulation they are from, and the part of the figure that is of interest? A reference link to where in the paper (section number) it is described and discussed is also required. I found myself flipping back and forward trying to work out what simulation was illustrated in which figure, please include more information to avoid this please. Line 5 to 15. I am assuming that this paragraph describes the physical phenomena that the viscous plastic rheology and associated yield curve are designed to replicate. However this

paragraph is worded such that it is an accepted and proved fact that sea-ice is viscous plastic and has behaviour that follows all these rules. Also the paragraph contains no citations. Rewriting this paragraph to emphasise that the viscous plastic rheology is designed to simulate the stress/strain relationship of sea ice over continuum length scales is required. It will also help to address the theoretical implications of using a rheology designed for the continuum approximation of sea-ice to simulate the deformation of a single floe.

Page 6 line 11 check parenthesis line 15 Is this 'theoretical angle' the one used to retrieve the LKF intersection angles from simulation results, and also with previous studies? If so can you state it here. The paper has not informed me how this study, and previous studies obtained the intersection angles widely discussed.

Page 8 line 6 is the model domain used in all experiments? If there are exceptions please list them. line 10 - 11 This statement about the robustness of the results is not backed up. Please refer the reader to the results that back up this statement. The proof of the robustness of this model needs to come first before any other results. Lines 11 - 17 Please state how the model time step works? I am not familiar with how the LSR solver for the VP rheology works. How does the model work in time? I am familiar with models that have a constant time step with solution iterations per time step. The model then continues time stepping for the required simulation period. Does your model work in the same way? Or is there a selected simulation time, and then the documented 1500/1500 iterations performed to cover the simulation time? If so then can you describe why the simulation time was selected as you have done with the spatial resolution. Is the simulation time and temporal resolution/number of iterations the same for all simulations? Results section A paragraph describing which simulations have been performed will be useful here. This will save having to flip back and forward through the paper to match results discussion and figures. The simulation 'robustness' results need to come first in order to validate the following results. line 29 Figure - figure (no capital), 'measured intersection angle' how was this angle measured? line

27, quantify 'right away' and how can a fracture appear but not in the deformation field? what field did it appear in? Can you comment on this time scale compared to observations of floe fracture (Dempsey et al. 2011 Fracture of a ridged multi-year Arctic sea ice floe )

Page 9 line 7 The 'robustness' results need to come first, then the model resolution and time period choice can be validated against them. line 9 extended time period, what was the original time period? line 10 total iterations or iterations per time step? what is the time step?

Page 10 Figure 3 Bottom left pane. Please use a bipolar colour scale for bipolar data. As in a different colour for +/-, white for zero for example. These scales are easily selectable. Line 8 'similarly as' - 'similar to' Section 3.3 If this section addresses the robustness then it needs to go first and also discuss the results in figure 5. Please also comment on the limitations of this model.

Page 11 Figure 5. First change or rescale the colourscale to highlight features. The max deformation appears to be around 10^-4, so limit the colour to this point, Also please label the colour scale legend with units. Why have you selected only 2 seconds of model run. what happens later in the run? Is this on a similar time scale to observations of floe fracture? Consider plotting a later time point if available or discuss how the model proceeds.

Page 12 again bipolar colour scale for bipolar field would be appreciated. All of your plotted fields so far have been for deformation, a plot of a stress field, if available, will be nice to see. There are lots of crack intersections in this plot. Is it possible to obtain all of these intersection angles? A distribution of angles could then be presented.

Page 13 Line 5 Comment on ice strength/thickness vs fracture angle. Is this a result you have observed? Or is it a theory that you are testing? Is there a citation for this theory? Line 17. You start this paragraph with statements about the link between initialised faults and deformation. Is this a theory you are testing? If so give a citation. Is it your

interpretation of the results? if so you need to state the results and the reference the figure first.

Page 14 Figure 8 caption is lacking detail. Is this the figure for the lateral confinement experiments, or ice thickness experiments? Please describe what is shown in every pane. Page 15 Figure 9, please label colour scale legends.

Page16 Figure 11, again please described the simulation this figure corresponds to? The colourscales are saturated so consider rescaling.

Page 17 line 18, A lot of LKFs - how many? Or is it more than compared to another simulation? rephrase or quantify. line 21 space before \theta

Page 18. Figure 12, are these Coulombic curve simulations? again this caption needs more detail. line 4 'realistic manner' how are they realistic? what simulations are you comparing to what observations? A figure reference and a citation are both needed for this statement.

Page 19 line 12 please refer to the section or figure or both that show the resolution and scale non-dependance. line 15 'appear' - 'appears' line 26 please give a citation for the statement on granular material.

Page 20 line 22 Citation required for 'Arctic-wide simulation' line 28 why is it unsurprising? Do mean that your results fit with previous theory and results? If so can you say this and cite them? line 33 'The ice open and create leads' - The simulated sea ice opens and creates leads

---

## Referee Comment (RC2) · Hutchings (Referee) · 17 Dec 2018

I am really pleased to see a paper that is making suggestions for idealised experiments we can use to differentiate between rheological models for sea ice. This in itself is worth publishing. The main result of the paper is that the elliptical rheology is inappropriate for representing observed cracking orientation in the ice pack, which is interesting and helps motivate changing sea ice rheological models used in climate and weather prediction.

I do have some concerns that the interpretation of observational data needs sharpening, and the results must not be overly interpreted given limitations of the use of RGPS data. Identifying intersection angles for lead pairs actually requires more work (and

dedicated field data collection) than this paper warrants. You are motivated by the fact that a simulation shows larger intersection angles than RGPS, and I agree that this is something to address. I just do not think you can use RGPS to determine what the intersection angle should be, just that it needs to be smaller. Note, there are others in the community funded to do this work of identifying fracture patterns associated with particular modes of failure. For example I have an NSF and NASA project that is looking at identification of modes of failure from satellite imagery. There are also upcoming field experiments that could provide case studies to constrain the actual behaviour of sea ice, which should provide further guidance for use of your idealised cases to constrain rheological model design. I would be happy to talk to you in person about using this analysis and data to support future model validation efforts. I would also caution you to be more careful in your description of the differences between VP and granular models. Some clarification missing from the manuscript is provided in my comments. I also have suggestions for why the VP model creates LKFs, which I feel is important for understanding the validity of LKFs in the viscous plastic sea ice model representing nature.

Specific Comments

Check spelling throughout the manuscript. Also check for missing brackets throughout. Grammar can be improved in places, and sometimes words are repeated. Make sure you have someone very carefully proof read the manuscript. I did not correct all the typos I saw because I am short on time and wanted to focus my attention on the central messages in your paper.

In the introduction be specific that you are considering conjugate fault pairs that form under specific confining stresses, the orientation of which is controlled by the yield curve shape and flow rule. In particular reference Pritchard in the introduction. It is only when I got to the conclusion that I saw you were aware of this work and it was motivating your study. It is wise to point out that not all applied stress will result in intersecting fault pairs (for example tension and pure compression do not).

Page 2 line 4. The appropriate references for efficient solution is Hutchings et al. 2004 or Jean-Francois Lemieux et al. 2010, I would not call LSOR or Hibler's method, which I used in the 2005 paper, as efficient. This introduces the efficient solution method that correctly couples P and U, for a convergent plastic solution. This solution method was not used in Hutchings et al. 2005. Hutchings et al. 2005 is the correct reference for qualitatively reproducing LKFs in the viscous plastic model.

Page 2 line 15: MEB? typo? I think you need to introduce the acronym for the Maxwell-Elastic-Brittle model here.

Page 2, line 24: argues -> argued.

Page 2, line 31: Flato and Hibler 1992 is not a mohr colomb relationship. The cavitating fluid behaves very differently and the first SIMIP (see work by Kryesher and Harder) indicated this was not a suitable stress-strain relationship for sea ice. Also check that Ip et al. 1991 is not using a different flow rule to Tremblay's. I am wondering if you are missing text here, as these two references were left hanging.

Page 2, some important points that I do not think are clear in your introduction:

The Elastic-Plastic model developed during AIDJEX was based on assumptions of a material with embedded cracks in all directions that are sub-grid scale. This is closer to a ductile material than granular material

The Viscous-Plastic model is only considered valid on course resolution (Hibler 1977). It is possible to consider this model with the ice always being in a state of plastic failure, until you get to high resolutions that allow representation of ice areas between fractures, when the viscous creep, while numerically small, is unphysical. At small scales an elastic model is appropriate for low stress states. The viscous behaviour inside the yield curve is often treated as regulaisation required for numerical solution.

Personally I think it is still not clear that the failure mode of a single floe is the same as an aggregate of floes. This has not been shown observationally or with models, and

statements of scale invariance based on observed qualitative correspondence between failure modes in the lab (cm scale) and ice pack (10-100km scale) do not extend to the floe scale.

Page 3 Discussion regarding orientation of intersecting LKFs from RGPS: I performed a similar analysis back in the early 2000's and never published the result, which was a wide spread in intersection angle. The reason I did not publish this is because I realised that the RGPS product could potentially be capturing fracture zones that form at different times, and therefore in the product appear to be a conjugate pair because they intersect, but they are not because they were not formed under the same confining stress. This is really obvious if you send some time on the ice pack in winter and observe leads forming and working. RGPS is not the right satellite product to use to identify conjugate fault pairs in sea ice. Hence I disagree that you can state "The wide range of intersection angles is presumably due to previous deformation history and associated hetrogeneity in the ice cover that dictates the strength locally".

page 3, line 32: Just want to clear up one very important point about my 2005 paper. It is steep stress gradients in the model sea ice stress field that allow LKFs to form. I suspect this opening is related to an instability in the model identified by Nico Gray (Gray and Kilworth 1995). We seeded stress gradients though a random number being added to P* (which defines compressive strength). At the time I wrote this paper I was obsessed with plastic convergence of the VP solution, so made sure there were no spurious stress values due to the numerical error. The pan-Arctic model of Heil's included in this paper, and other VP models, are able to show LFS because of the noise introduced by not converging fully to the yield curve. If you play around with a VP model you can create divergence related instabilities along gradients in forcing (e.g. non-smoothly interpolated wind fields), or even have the model blow up and crash due to one localised discontinuity in thickness (e.g. I have seen this when using a nudging method to assimilate data into the CICE model that created open water locally). The reason I bring this up is that I feel it is very important that people understand how the

VP model can create LKFs. The mechanism is quite different from what might actually be happening in a granular or brittle material. I would be very happy to advise on experimental design, repeating and following up on my investigations 15 years ago.

page 3, line 34: The original study on shape of yield curve and ice arches is in Billy Ip's thesis, that was published later by Hibler in Hibler et al. (2006).

page 8, line 14. A reader unfamiliar with numerical solution of the VP model will need some guidance as to what non-linear and linear iterations are. I know you are talking about the sub-cycling to reach plastic equilibrium (or close to it) and converge the velocity solution at each time step. Perhaps use language that is more obvious to a casual reader. Incidentally, did you check convergence properties? Just curious. I think you point out somewhere that the modified coulombic rheology is slower to converge - I found that solutions for yield curves with corners never converged fully. A frustrating reality! If you follow my suggestions to delve into why the model creates LKFs you will need a full description of the interative process and convergence characteristics.

page 8, results section: Describe what the applied strains are in the numerical experiments (magnitude, not just direction).

page 8 line 26: What are the default parameters? I think you forgot to reference table 1.

page 9, line 4: Regarding your statement "Fracture occurs when the stress state intersects the yield curve". Plastic failure occurs then. The fact that a "fractures" form is because the ice deforms at a stress discontinuity where the stress accumulates and reaches yield. You are correct in pointing out that the strain-rate has characteristic directions along which divergence will occur, defined by the shape of the yield curve and flow rule. It is this divergence, relative to the confining stress, that defines the directions of the linear deformation features in the model runs.

Comment on differences between 3.1 and 3.2: The change in nature of cracks when

you decrease the number of internal iterations (linear iterations) is probably related to the fact that the ice stress field is more heterogeneous (further from the converged solution) and the LSOR method tends to create noise in the stress field and smoothes with increasing number of iterations (unlike the SIMON method I proposed, Hutchings et al. 2004, that has a smoother convergence to the yield curve). Hence there are more points where LKFs can nucleate when you reduce the number of internal iterations. This is just a suggestion, with out looking at the stress fields in your experiments I can not tell you if this is what is actually happening. Incidentally, another unpublished result that I presented at AGU in 2003: The VP model can create intersecting deformation features across the entire Arctic Ocean is you do not converge to plastic equilibrium and are not careful in smoothing the solution between time steps (which can be done numerically through the choice of advection scheme or Bill's introduction of artificial diffusion in his 1979 paper). I never followed up this work. I suspect that this is a direct consequence of Gray's instability. This instability is damped by the addition of numerical diffusion (or artificial diffusion) in the solution proceedure. We might think the resultant strain-rate fields are more realistic, I just do not believe using the non-convergence and numerical instability is an appropriate way to model the process because we are not controlling the nature of the stress concentrators or stress propagation in the model appropriately. The key point is that the ice pack strength is highly heterogeneous and while we do not know the nature of the stress concentrators in the ice pack, they are likely to be more randomly distributed (which non-convergence to the yield curve might be approximating, but is not controlled for). There is a need to understand the nature and distribution of the stress concentrators in sea ice, so we can appropriately model this. And it would really help future sea ice modellers to point out this issue more clearly in papers that investigate LKFs in the viscous-plastic model.

Figure 7: Nice illustration of the role of boundary conditions on the stress solution. Section 3.5: Good illustration. I would suggest you critically look at the stress fields in your previous experiments to identify what the stress concentrators are there. Did you forget to reference figure 9 in this section. Finally, Bill Hibler has shown similar

results where embedded fractures of different orientations would join together to form larger scale fracture patterns. Not sure he published that, but I think he did. Look at the papers he wrote with Aksenov and his first anisotropic paper with embedded leads in grid cells. Unfortunately I am on an airplane right now and don't have access to his papers.

page 16 line 1: Here and in other places you confuse the simulation with reality. "This is in contrast with other granular materials". Remove "other", as this is in contrast with granular materials. The VP model is not modelling a granular material. While sea ice may be a granular material, the rheology is designed for different behaviour.

page 16 line 5: "larger that what" -> "larger than that"

page 16, line 9: I would like to see the strain-rate field for the longer simulation with e=0.7 where deformation is in convergence. Where in the field is the ridging occurring? What do the intersection angles look like?

page 17 line 2: "and individual floes form" could be clarified as "4 separate floes form".

page 17 line 15: Please clarify the statement "the fracture pattern is very sensitive to coefficient of internal friction. This makes measuring the fracture angle very difficult." Surely the sensitivity will help you differentiate fracture angles. Incidentally mu was not defined in section 2 or here.

What causes the spread in the stress state? Is this related to the opening/ridging and subsequent ice strength changes? So the spread in stress state is controlled by the strength parameterisation. I feel this is important to point out, because it is another control we have on the spread of intersection angles you might see under a particular confining stress.

page 17 line 25: Clumsy language: "the stress state touches the yield curve on both parts of the yield curve." Very unclear that you mean the stress state falls on the coulombic limb and the ellipse cap. rephrase.

page 19 top paragraph: This is a matter of opinion. I disagree that the ice pack is characterised by diamond shaped floes. Yes, diamond floes form under certain confining stresses, but is this the most prevalent mode of failure in winter? That needs to be proven. This point does not discount your use of your numerical experiments to differentiate between rheologies, but it does question if an anisotropic rheology based on diamond shaped floes is appropriate for all space and time.

Also, did you calculate characteristic directions for the VP model to confirm these are controlling the diamond structures in your simulations? I have a code somewhere (from 15 years ago) that does this. If I can find it I can give it to you.

page 19 line 12: This sentence is miss-representative: "Thus, the rheology is shown to be scale independent ... in line with observations". Your numerical experiment is set up to ensure the behaviour is scale independent from the scale of the grid size to the domain size. You would really hope your numerical experiment results do not depend on resolution (good practice to check this) and there is no reason scale should change intersection angles for the reasons you have stated previously. Rephrase this statement so someone does not quote it as evidence supporting Schulson's hypothesis.

page 19 line 20: Just to clarify, the reason the experiments with thin ice change the fracture angle is because the presence of the thin ice modifies the stress state across the domain. So with an ellipse this will change the intersection angle, with a coulombic rheology it would not. I feel this part of the paragraph needs more clarification.

page 19 line 22: Your interpretation of the RGPS data (if one believes the intersection angles are at conjugate pairs and not leads formed at separate times) would lead one to believe that there is not a constant fracture angle independent of confining stress.

page 20 line 2: Perhaps clarify that the Miller et al. (2005) experiments were using metrics of ice thickness, area and velocity to determine the optimal yield curve shape. It is my memory they did not consider the form of the ice strength parameterisation as an alternative to changing shear strength, or yield curve shape, just the eccentricity of

the ellipse.

page 20 line 5: Can you show that your numerical experiments are consistent with Pritchard (1988).

page 20 line 8: questions -> questioned

After reading your discussion I wondered if the tear drop yield curve (originally proposed by Pritchard) might be more appropriate that the Hibler modified ellipse / mohr coulomb.

Also, Hibler and others recognise that you must have a closed cap on a coulombic rheology to allow ridging. Perhaps the ellipse is not the best choice for this. In engineering it is more common to have a flatter closure to the yield curve.

page 20 line 26: sensible -> sensitive

page 20 line 26: Another example where it is not so clear you are talking about the VP rheology problem: "The fracture angles is also sensitive to the surrounding sea ice cover, in contradiction to the granular nature of sea ice". Also, the stress field is going to depend on surrounding ice even when the ice is modelled as granular, which I am not sure is what you were meaning to imply is not true for granular materials. I think you need to clarify the language, and I think I disagree with you that this test is suggesting ice is granular - it would be something we could test in an ice tank to find out what the actual behaviour is though.

I feel you do not highlight a key result in the paper: That fracture angles below 30o are not possible with the elliptical rheology, and that this is in direct conflict with observational evidence for smaller fracture angles. Even in light of errors of interpretation of the RGPS intersection angles this result still holds.

page 21 line 3: Note that at cusps in a yield curve two possible solutions are possible. I feel you can clarify your point about not using non-differentiable yield curves. They are also numerically unstable. The unclear fracture pattern is not something I have issue

with. Perhaps this exists in reality when the stress state can spread across opening and closing modes.

page 21 line 10: I feel you can be stronger here in stating that the ellipse with normal flow rule can be discounted as unphysical.

page 21 line 11: Scale is really unimportant in these experiments. You can perform them on any scale. The more important question is what scale do these types of fracture events actually occur on and can that be resolved in models?

I see you did not reference work by K. Wang (2007) who used lead intersection angles to try to estimate the shape of a yield curve. He also has a paper were he performed a similar study to you (Wang and Wang 2010), however for the pan-arctic and perhaps with convergence issues that make his findings hard to interprete. While this work suffered from problems of representativeness of the observational data (how can you be sure fractures formed at the same time), as you do, I feel you should consider Wang's papers in light of your findings.

Finally, I believe that the stress state between fractures in your numerical experiments is inside the yield curve (viscous), and the motion close to zero. Is this correct, it was what I found when I was working on this. Just a point to clarify that the accumulation of stress along fractures is due to the yield curve discontinuity, and the associated characteristic directions in the strain field that control the propagation of fracture direction. This accumulation of stress needs to be nucleated at a location with high stress gradient (such as a corner on the boundary or strength/stress difference between grid cells). Once the stress reaches the yield curve, the numerical instability is probably put into play during the inner iterations. You do not see LKFs in VP models that have smooth boundaries and strength fields. The formation of LKFs is grid resolution dependent (as the linear instability identified by Gray is). You have speculated on why LKFs form in the VP model only at higher resolutions in a previous paper and I would suggest the place to look is in the convergence of the solver, and the splitting of velocity solution

from the ice strength (pressure). I do not think it is just the fact that divergence (and strength reduction) can be greater at higher resolution. Clarifying this mechanism will help readers understand why VP models show this behaviour. It will also hopefully get people thinking about how to represent stress accumulators in the model, because many people using the VP model and studying fractures are unaware of how the model produces these.

Jenny

References

Gray and Kilworth (1995) Stability of the viscous plastic sea ice rheology. J. Phys. Ocean. 25(5), 971-978.

J-F. Lemeuix, B. Tremblay, J. Sedlacek, P. Tupper, S. Thomas, D. Huard, J-P. Auclair. 2010. Improving the numerical convergence of viscous-plastic sea ice models with the Jacobian-free Newton-Krylov method. J. Comp. Phys., 2840-2852.

Hutchings J.K., H. Jasak and S. W. Laxon 2004. A strength implicit correction scheme for the viscous-plastic sea ice model. Ocean Modelling.

Hibler W.D. III, J.K. Hutchings, and C. F. Ip, 2006. Sea-ice arching and multiple flow state of Arctic pack ice. Ann. Glaciol. 44.

Hibler W.D. III. 1977. A viscous sea ice law as a stocahastic average of plasticity.

Wang, K. 2007, Observing the yield curve of compacted pack ice, J. Geophys. Res. 112, C05015.

Wang, K. and Wang, C. 2010, Modelling linear kinematic features in pack ice, J. Geophys. Res. 114, C12

---

## Author Comment (AC1) · 14 Feb 2019

**Answer to tc-2018-192-RC1 – Harry Heorton**

February 14, 2019

**Note:**

- The referees comments are shown in black.

- **The authors answers are shown in bold typeface and colored in blue.**

- **The modifications brought to the manuscript are shown in bold typeface and colored in gray.**

**Global comments**

**R1#1,** Modelling Sea Ice fracture at very high resolution with VP rheologies This paper documents idealised very high resolution (25 m) numerical simulations of deformation in sea ice using the viscous plastic VP rheology. The sea ice is predominantly put under uniaxial compression from the top and bottom boundaries and the resulting deformation features are documented. Simulations are performed with longer model run times, decreased spatial resolution, modified boundary conditions, biaxial compression, imposed flaws within the sea ice and alternate rheologies by changing the yield curve shape. The results are well documented, though further detail is required on the model setup. Particular emphasis is put on the resulting deformation or linear kinematic feature intersection angle, as a means to provide a link between simulated and observed deformation features, and thus provides insight into how to select an appropriate sea ice rheology for simulation within climate models or for future studies. This method presents an exciting way to link observations and simulations of sea ice deformation, but there are certain aspects of the study that, in my opinion, need addressing.

**We thank the reviewer for a thorough review and for highlighting grammatical mistakes in the original manuscript**

**R1#2,** Firstly is selection of model resolution, and consideration of whether to simulate a single floe or a continuum of many floes. The selected 25 m resolution makes this a simulation of a single floe, and the paper describes the study as such, but there is little discussion of whether the selected VP rheology is a valid representation of a single floe. The VP rheology was developed to simulate a continuum of floes over ocean length scales, and it is not immediately obvious to me, that the VP rheology is a valid representation of the deformation of a single floe just by reducing the model resolution to that of single floe, and setting the ice concentration to unity as performed in this study. I am unaware of previous simulations of solid body deformation from other fields, and it may be the case that VP like rheologies are well studied, but this needs to be discussed in this paper. On the other hand this paper may be be a proof of concept that the VP rheology can be

used to simulate the deformation of a single floe, and if it is, then it needs to be worded as such.

Typical floe size in the Arctic Ocean are of the order of 10 km, so that the continuum assumption can only be valid for model spatial resolution of 100km (Overland et al., 1995). Despite this, the use of VP rheologies with spatial resolution lower than 100km is now common practice even in lower resolution Global Climate Models. This raises the valid concern expressed by the reviewer. We argue that if the modes of failure in a single ice floe are the same as those of an aggregate of floes, then the continuum assumption can be used at spatial resolution higher than the 100km barrier. This is the implicit assumption currently made in the community with the use of higher spatial resolutions even in lower resolution Global Climate Models. We note that the mode of fracture in continuum models is independent of the scale of the problem. For instance, the results presented in the paper would not change if the domain and spatial resolution were increased by a factor of 500 (i.e. dx = 125km).

We added on page 2 of the revised manuscript :”*It can be argued that if the mode of deformation of a single floe is similar to that of an aggregate of floes, a given rheology developed for a continuum can still be applicable at spatial resolutions of the order of the floe size (Overland et al., 1998), but the validity of a given flow rule across scales is not clear.*”

**R1#3,** A further implication of simulating a single floe, rather than a continuum floes, is the observations that the paper discusses when validating the results. The RADARSAT and RGPS data is given as a means to validate the yield curve selected. But the RGPS data is given on continuum length scales, so to validate a selected yield curve the comparison will need to be made with simulations over continuum length scales, not on the floe length scales presented in this paper. To validate the simulations of this paper observations of individual floe shape, aspect ratio and floe-to-floe crack intersection angle from aerial images of floes will be best suited. Secondly the presented results are often from the order of 1-10 seconds into the simulations. This is in contrast to idealised deformation experiments of Hutchings et al. (2004) and Heorton et al. (2018) where the results are given at 1 - 24 hours of simulation, and explored over multiple days. Within these studies the initial conditions and early stress/deformation states are explored and documented. Also in both of these cases, the initial state of the ice over was seeded with noise (in strength, or thickness and forcing) to allow for features to develop and stop an unrealistic uniform sea ice cover. Were such considerations performed in this study? Are the initial conditions uniform? What are the implications of using results from 1-10 seconds of model run? I would like to see documentation of the time series of stress/strain state in order to validate the idealised experiments and the initial conditions. I personally found the longer documented run of 45 minutes interesting, with individual floe-like shapes appearing that can be compared to images of floes.

- We did not use observations of angles of fracture from RADARSAT and/or RPGS dataset to validate the model results. Instead, we use a comparison study by Hutter et al. (2018b) between observed (RGPS) and simulated Linear Kinematic Features as a motivation for the present work. As mentioned above, the results presented are independent of spatial resolution, and the conclusions are applicable to continuum scale observations. This shown in Section 3.2.1 and on Figure 5.

- The goal of the study is document the fracture angle as a function of mechanical strength parameters, boundary conditions, and to compare the rheology-dependent model of deformation from those induced by inhomogeneities in the initial thickness field (Fig. 9) as in Heorton et al. (2018), and Hutchings et al. (2005). The fracture angle provides, for the first time, a meaningful diagnostic (since it is related to deformation and motion of sea ice) that allows a discrimination between different rheologies (yield curve and flow rule). Note that prior attempts using sea ice drift and PDF of sea ice deformation did not allow for this discrimination (Kreyscher et al., 1997; Bouchat and Tremblay, 2017; Hutter et al., 2018a). For this reason, we only integrate the model for a few seconds - until the fracture is apparent in the thickness or deformation field. Results from longer simulations were also performed to demonstrate that VP rheologies can simulate secondary fracture lines similar to observations with lead opening and ridging (Fig 7).

- The initial conditions are uniform in ice concentration and ice thicknesses. This has been clarified in Section 2.3 of the revised manuscript. Ice strength was not modified.

- As it is now explained in section 3.1 on page 9 of the revised manuscript, the ice floe fails starting from the first time step with stress states on the edge of the yield curve.

**R1#4,** Also there is little discussion of the implications and 'robustness' of the presented model results. The authors described the results as 'robust' on multiple occasions but make little effort to inform the reader why they are robust. Figure 5 is presented as a domain/resolution study but only mentioned once in passing as an indication of the models robustness. I'm assuming that the model domain/resolution/run time has been investigated but there is no documentation or discussion of the models limitations. A section describing this is required in order to allow the other results to be published.

As stated above, we are not interested in subsequent deformation after ice fracture. For this reason, we do not present results on the sensitivity to time of integration. We changed the organisation of the results section of the manuscript by regrouping all sensitivity experiments in one section (3.2), including a short description on the sensitivity of the results to spatial resolution and domain size on page 11, Section 3.2.1, of the revised manuscript.

**R1#5,** Thirdly it is not obvious how the deformation or linear kinematic feature intersection angle were calculated. I would like to see this information given in an appendix, such a method is a very useful contribution to studies of sea-ice rheology and one that I would like to use in the future. A citation to another study where this method was performed and is described is another option. The given appendix showing the theory behind internal friction and Mohr-Coloumb failure criterion, whilst interesting to see, does not appear to be original theory for this paper and is not required and the dependencies can be stated and cited.

- The angle of fracture is measured using a free image processing software (GNU Image Manipulation Program, GIMP). The small number images to treat did not call for a special program to measure angles as done in Linow and Dierking (2017), Mohammadi-Aragh et al. (2018) and Hutter et al. (2018b) was not necessary because the number images to be processed was small. **To clarify this issue, we**

added the following text on page **7** of the revised manuscript: *"The angle of fracture is measured with the angle measuring tool of the GNU Image Manipulation Program (GIMP, https://www.gimp.org/). A special automatic algorithm to measure angles is described in Linow and Dierking (2017); Hutter et al. (2018b).".*

- We decided to keep the Mohr-Coulomb theory in Appendix A for the sake of completeness. We have added the following reference in Appendix A for readers who want to see a related description of the theory (Hibler and Schulson (2000) Pritchard (1988).

**Specific comments:**

**R1#6, Page 1 Line 6-8** What are the dependencies of typical granular materials? The sand castle analogy is not useful.

We decided to remove the sand castle analogy from the abstract but to keep it in the discussion/conclusion because it is something that anyone within or outside the community is familiar with; it was also well received at the future of Earth System Modeling workshop at CalTech in November 2018. We rewrote the abstract.

The new abstract is *"Recent high resolution pan-Arctic sea ice simulations show fracture patterns (Linear Kinematic Features or LKFs) that are typical of granular materials, but with wider fracture angles than those observed in high-resolution satellite images. Motivated by this, ice fracture is investigated in a simple uni-axial loading test using two different Viscous-Plastic (VP) rheologies: one with an elliptical yield curve and a normal flow rule, and one with a Coulombic yield curve and a normal flow rule that applies only to the elliptical cap. With the standard VP rheology, it is not possible to simulate fracture angles smaller than $30°$. Further, the standard VP-model is not consistent with the behaviour of granular material such as sea ice, because: (1) the fracture angle increases with ice shear strength; (2) the divergence along the fracture lines (or LKFs) is uniquely defined by the shear strength of the material with divergence for high shear strength and convergent with low shear strength; (3) the angle of fracture depends on the confining pressure with more convergence as the confining pressure increases. This behavior of the VP model is connected to the convexity of the yield curve together with use of a normal flow rule. In the Coulombic model, the angle of fracture is smaller ($\theta = 23°$) and grossly consistent with observations. The solution, however, is unstable when the compressive stress is too large because of non-differentiable corners between the straight limbs of the Coulombic yield curve and the elliptical cap. The results suggest that, although at first sight the large scale patterns of LKFs simulated with a VP sea ice model appear to be realistic, the elliptical yield curve with a normal flow rule is not consistent with the notion of sea ice as a pressure-sensitive and dilatant granular material."*

**R1#7, Page 1, Line 8** what model? this paper or previous?

We meant the mathematical description of the ice, the rheology, i.e. the elliptical hibler Viscous-Plastic rheology implemented in MITgcm.

The abstract was changed (see R1#6 above), this sentence has been deleted.

**R1#8, Page 1, Line 10-14** More description of 'typical granular materials' that are not accurately described and all comparisons are difficult to follow. I would avoid these loose comparisons in the abstract and stick to definite results.

We wanted to give the reader a hint of the significance of our results by giving a comparison to granular material properties. But it might lead to confusion. We will limit the comparison to the discussion part of the paper.

We modified the abstract as specified above for comment R1#6

**R1#9, Page 1, Line 24** are the two citations model studies or observations of sea ice floes? Observational studies are required for this sentence.

Overland et al. (1998) is an observation study base on buoy data, while Tremblay and Mysak (1997) is a modeling study. We remove this last reference and replace it by another observational study (Rothrock and Thorndike, 1984).

**R1#10, Page 2 Line 5** Leads plural, dangerous to use the word memory when describing a computer model, 'emergent anisotropy' is more accurate.

Both have been corrected as suggested.

**R1#11, Page 2, Line 8** The equations are difficult to solve due to their non-linearity and complexity not because of sea ice. It is however difficult to represent sea ice with simple, easily solvable equations due to it's non-continuous features.

We changed this sentence to express our thought more clearly.

The first sentence of the 2nd paragraph of the introduction is changed to : *"The sea ice dynamics are complicated because of sharp spatial changes in material properties associated with discontinuities (e.g. along sea ice leads or ridges) and heterogeneity (spatially varying ice thickness and concentration). The sea ice momentum equations are difficult to solve numerically because of the non-linear sea ice rheology."*

**R1#12, Page 3 Line 24** argues - argued
Corrected as suggested.

**R1#13, Page 3 Line 28** rheologies plural
Corrected as suggested.

**R1#14, Page 3 Line 32** check citations and parenthesis
Corrected as suggested.

**R1#15, Page 3 Line 33** check citations and parenthesis
Corrected as suggested.

**R1#16, Page 3 Lines 34-35** 'Based on these satellite observations, amongst others (provide some examples), and in-situ...'

The sentence was re-written as: *"Based on satellite observations (e.g. RADARSAT Geophysical Processor System, RPGS, or Advanced Very-High-Resolution Radiometer, AVHRR), and in-situ internal ice stress measurements (e.g. from the Surface Heat Budget of the Arctic Ocean, SHEBA,*

*experiment),[. . . ]"*

**R1#17, Page 3 line 6** space before 'Girard'
Corrected as suggested.

**R1#18, Page 3 line 13** delete parenthesis
Corrected as suggested.

**R1#19, Page 3 line 15** 'are appear as line of' - 'appear as lines of', deformation singular, 'with the deformation' - 'with shear deformation', divergence - convergence
Corrected as suggested.

**R1#20, Page 3 Line 18** leads - leading
Corrected as suggested.

**R1#21, Page 3 line 19-20** check citation parenthesis
Corrected as suggested.

**R1#22, Page 4 line 1** Wilchinsky et al. also deduced intersection angles between floes that are relevant to this paper.
The reference to Wilchinsky et al. (2010) has been added to the list of citation.

**R1#23, Page 5 Figure 1**. What is 'Mohr-Coloumb flat' In general I found the figure captions to be lacking in content. Can they all be expanded to directly describe what simulation they are from, and the part of the figure that is of interest? A reference link to where in the paper (section number) it is described and discussed is also required. I found myself flipping back and forward trying to work out what simulation was illustrated in which figure, please include more information to avoid this please.
We re-wrote the figure captions and made them self-sufficient as suggested by the reviewer. *Flat* refers to the linear part of the yield curve. We replaced the word *flat* by *linear limb* which is more accurate.
The captions of all figure have been extended to be self-sufficient. The caption of figure 5 now reads *"Elliptical yield curve (black) with ellipse aspect ratio $e = a/b = 2$. Coulombic yield curve (red) and elliptical capping with internal angle of friction ($\mu$). Both $e$ and $\mu$ are measures of the shear strength of the material. The normal flow rule applies only to the elliptical part of the yield curves. For the two straight limbs of the Coulombic yield curve, the flow is normal to the truncated ellipse (dash-dot line) with the same first stress invariant. Note that the axes $\sigma_1$, $\sigma_2$ and $\sigma_I$, $\sigma_{II}$ do not have the same scale."*

**R1#24, page 5 Line 5 to 15**. I am assuming that this paragraph describes the physical phenomena that the viscous plastic rheology and associated yield curve are designed to replicate. However this paragraph is worded such that it is an accepted and proved fact that sea-ice is viscous plastic and has behaviour that follows all these rules. Also the paragraph contains no citations. Rewriting this paragraph to emphasise that the viscous plastic rheology is designed to simulate the stress/strain relationship of sea ice over continuum length scales is required. It will also help to address the theoretical implications of using a rheology designed for the continuum approximation of sea-ice to

simulate the deformation of a single floe.

*This paragraph was reworded taking into account the comment from the reviewer. Please see page 6 of the revised manuscript.*

*This paragraph now starts with "The VP rheology was originally developed to simulate ice motion on a basin scale (e.g., Arctic Ocean, Southern Ocean) (Hibler, 1979). In this model, stochastic elastic deformation is parameterized as highly viscous (creep) flow (Hibler, 1977). Ice is set in motion by surface air and basal ocean stresses moderated by internal ice stress."*

**Page 6 line 11** check parenthesis

*The missing parenthesis have been added*

**R1#25, Page 6 line 15** Is this 'theoretical angle' the one used to retrieve the LKF intersection angles from simulation results, and also with previous studies? If so can you state it here. The paper has not informed me how this study, and previous studies obtained the intersection angles widely discussed.

*Yes it the theoretical angle of fraction derived form the Mohr's circle and the Mohr-Coulomb yield criterion. It is described in several paper, e.g. in Ip et al. (1991); Hibler and Schulson (2000), and , as shown in appendix A and B, it is in agreement with the characteristics lines described in Pritchard (1988)*

*The 3 references above have been added on page 8 of the revised manuscript in the sentence : "The theoretical angle of fracture $\theta$ can be calculated from the Mohr's circle of stress and yield curve written in the local (reference) coordinate system (Ip et al., 1991; Pritchard, 1988; Hibler and Schulson, 2000). Details are described in the appendix. For a Mohr-Coulomb yield criterion, $\theta$ follows immediately from the internal angle of friction, that is the available shear strength. An instructive analogue is the slope of a pile of sand on a table. Wet sand can support more shear stress and hence the slope angle can be steeper (smaller)."*

**R1#26, Page 8 line 6** is the model domain used in all experiments? If there are exceptions please list them.

*This model domain is used in all experiments, except for two experiments reported in Section 3.2.1 (Figure 5 and Figure 7).*

*We inserted : "The model domain is a rectangle of size $10\,km \times 25\,km$, except for Sect. 3.2.1 and Sect. 3.2.2" on page 8 on revised manuscript*

**R1#27, Page 8 line 10 - 11** This statement about the robustness of the results is not backed up. Please refer the reader to the results that back up this statement. The proof of the robustness of this model needs to come first before any other results.

*See Section 3.2 on page 10 of the revised manuscript for a discussion on the sensitivity of the results to different boundary conditions. In our opinion, the demonstration that VP rheologies can simulate realistic fracture lines that have angles in accord with theory should be presented first. This is the reason why we present these results first, before the sensitivity of the results to the boundary conditions. The boundary conditions in this context can be seen as external forcing on the interior solution.*

**R1#28, Page 8 Lines 11 - 17** Please state how the model time step works? I am not familiar with how the LSR solver for the VP rheology works. How does the model work in time? I am familiar with models that have a constant time step with solution iterations

per time step. The model then continues time stepping for the required simulation period. Does your model work in the same way? Or is there a selected simulation time, and then the documented 1500/1500 iterations performed to cover the simulation time? If so then can you describe why the simulation time was selected as you have done with the spatial resolution. Is the simulation time and temporal resolution/number of iterations the same for all simulations?

The non-linear momentum equation is solved iteratively until a converged solution is obtained. Typically 1500 iterations are required to reach convergence. Then the external forcing is then updated and a new solution calculated. This has been clarified on pages 7-8 of the revised manuscript.

We modified the text describing the numerical solver :"*We solve the non-linear sea-ice momentum equations with a Picard or fixed point iteration with 1500 non-linear or outer-loop (OL) iterations. Within each non-linear iteration, the non-linear coefficients (drag coefficients and viscosities) are updated and a linearized system of equations is solved with a Line Successive (over-)Relaxation (LSR) (Zhang and Hibler, 1997). The linear iteration is stopped when the maximum increment is less than $\epsilon_{LSR} = 10^{-11}\, m\, s^{-1}$, but we also limit the number iterations to 1500. Typically, 1500 non-linear iterations are required to reach a converged solution. This is so because of slow convergence due to the highly non-linear rheology term and the high spatial resolution (Lemieux and Tremblay, 2009).*"

**R1#29, Results section** A paragraph describing which simulations have been performed will be useful here. This will save having to flip back and forward through the paper to match results discussion and figures. The simulation 'robustness' results need to come first in order to validate the following results.

A paragraph have been added at the beginning of the Section 3 (Results) on page 9 of the revised manuscript, stating the different part of the result section. A paragraph has been added at the beginning of Section 3.2 on page 10 of the new manuscript to list the sensitivity experiment that have been performed.

The first paragraph of Section 3.2 now reads "*We use simple uni-axial loading experiments to investigate the creation of pair of conjugate faults and their intersection angle. After presenting the results of simulations with the default parameters (Section 3.1), we explore the effects of experimental choices: confining pressure, choice of boundary conditions (i.e. von Neumann versus Dirichlet), domain size and spatial resolution and inhomogeneities (i.e. localized weakness) in the initial thickness and concentration field (Section 3.2). Finally, we study the behaviour of two viscous plastic rheologies with different yield curves and compare these dependencies to what we can infer from smaller and larger scale measurements from laboratory experiment and RGPS observations (Section 3.3).*"

**R1#30, Page 8 line 29** Figure - figure (no capital), 'measured intersection angle' how was this angle measured?

We modified the manuscript to comply with the journal standards, i.e. using *Fig. or Figure, Eq. or Equation , Table, and Sect. or Section.*

The word *measured* is now replaced by *simulated*. We measured the angle using the GIMP software. See response to comment R1#5 above for more details.

**R1#31, Page 8 line 27** quantify 'right away' and how can a fracture appear but not in the deformation field? what field did it appear in? Can you comment on this time scale compared to observations of floe fracture (Dempsey et al. 2011 Fracture of a ridged multi-year Arctic sea ice floe )

After 1 timestep, the stress states reach the yield curve and deformation occurs. We see this immediately in the strain rates (divergence and deformation). For the results presented in the paper, we have iterated for 10 additional seconds in order for the signal to also be seen in the thickness and concentration field. We do this to more clearly show the fixed link between sea ice shear strength and divergence in the standard VP rheology of Hibler. We removed the sloppy term "right away" from the text. The reference to Dempsey et al. (2012) was also added in the discussion.

The new text on page 9 of the revised manuscript now reads: : *"After 1 timestep (or 0.1 s), the stress states already lie on the yield curve and the fracture is readily seen in the deformation fields (divergence and shear). We iterate for a total of 20 seconds in order for the signal to be apparent in the thickness and concentration fields. We do this to more clearly show the link between position of the stress states on the yield curve and the normal flow rule in the standard VP rheology of Hibler (1979)."*

On page 22 of the revised manuscript (discussion section), we added the following sentence : *"Observed time scales of fracture are on the order of 10 seconds for 60 m floe diameters (Dempsey et al., 2012, Figure 6 top right panel) and from typical elastic wave speeds of 200–2000 $m\,s^{-1}$, large cracks of order 1000 km can form in minutes to hours (Marsan et al., 2012)."*

**R1#32, Page 9 line 7** The 'robustness' results need to come first, then the model resolution and time period choice can be validated against them.

Please see response to comment **R1#27** above for a justification.
*The angle of intersection between a pair of conjugate fault does not change with domain size and spatial resolution (see Fig. 5).*

**R1#33, page 9 line 9** extended time period, what was the original time period?

The original time period of the simulation is 20 seconds with a 0.1 second time step. The length of the simulations is not important here as we are showing only results from the first timesteps.

We added a reference to Table 1 in the revised manuscript on page 9 section 3.1. We also changed the sentence on page 7 (please see next comment R1#34 below)

**R1#34, Page 9 line 10** total iterations or iterations per time step? what is the time step?

It is the number of sub-cycles used to solve the non-linear momentum equation. This should be clearer in the new version of the manuscript (see response to comment R1#28 above)

The first two sentences now read: *"Continuing the integration to 2700 seconds (45 min), compared to 20 seconds in the reference simulation leads to the creation of smaller diamond-shaped ice floes due to secondary and tertiary fracture lines (Figure 6)."*

**R1#35, Page 10 Figure 3** Bottom left pane. Please use a bipolar colour scale for

bipolar data. As in a different colour for +/-, white for zero for example. These scales are easily selectable.

The colorbar were changed as suggested.

**R1#36, Page 10 Line 8** 'similarly as' - 'similar to'
Corrected as suggested.

**R1#37, Section 3.3** If this section addresses the robustness then it needs to go first and also discuss the results in figure 5. Please also comment on the limitations of this model.

Please see the new Section 3.2 and the answer to the comment R1#27 above.

The limitations of the VP model are discussed in the introduction on page 2 of the revised manuscript. All sensitivity experiments - including sensitivity to spatial resolution and domain size - are now presented together in a new section 3.2 entitled: *Sensitivity experiments.* The limitations of the this study are discusses in the discussion section on page 22 of the revised manuscript.

We added in the discussion section on page 22 :*"The simulations presented in this study are not realistic and cannot be compared directly to observations of ice floe fracture. For instance, our idealized ice floe is homogeneous while sea ice is known to feature some weaknesses like thermal cracks or melt ponds."*

**R1#38, Page 11 Figure 5** First change or rescale the colourscale to highlight features. The max deformation appears to be around $10^{-4}$, so limit the colour to this point, Also please label the colour scale legend with units. Why have you selected only 2 seconds of model run. what happens later in the run? Is this on a similar time scale to observations of floe fracture? Consider plotting a later time point if available or discuss how the model proceeds.

We want to keep the same scale for every plot. Yes, the maximum value is approximately equal to $10^{-4}$. We added color bar legends with units. We limited the simulation to 2 seconds because of computational constraints. After 2 sec, the fracture angle is already visible, and it is not necessary to run the simulation any longer time, because this would simply make the signal stronger in the thickness and concentration fields. This was tested using a smaller domain. We clarified this on page 11 of the revised manuscript.

We have re-written the captions of all figures including the reference to the appropriate section as suggested by the reviewer. We justified our choice of total integration of 2 sec for this experiment in the revised manuscript.

The caption was modified : *"Maximum shear strain rate (second strain invariant) after 10 seconds of integration for the default domain size and $\triangle x = 100\,m$ (a) and $500\,m$ (b), and for the default $\triangle x$ and a doubled domain size of $20\,km \times 50\,km$ (c). Note that for case of the double domain (c), the southward velocity at the northern boundary was also doubled to keep the deformation rate constant, and that this simulation is limited to 2 seconds for numerical efficiency."*

**R1#39, Page 12** again bipolar colour scale for bipolar field would be appreciated. All of your plotted fields so far have been for deformation, a plot of a stress field, if available, will be nice to see. There are lots of crack intersections in this plot. Is it

possible to obtain all of these intersection angles? A distribution of angles could then be presented.

While these simulations are indeed cool to look at (especially when you animate them!), it does not seem useful to us to investigate the angles of all of these leads in detail. Most of them were created after the initial fracture, therefore the direction of stress and magnitude of stress have been modified by ensuing fractures and deformations so that the analysis would be confounded. We can see that the fracture pattern is not absolutely symmetrical. This means that the converged solution is not reached. In principle, this is possible, e.g. with the software of Hutter et al. (2018b). We find the stress field not to be helpful in our case, the deformation field showing the fracture lines is more important for us to explore the effects of the rheology.

The colorbar has been corrected as suggested.

**R1#40, Page 13 Line 5** Comment on ice strength/thickness vs fracture angle. Is this a result you have observed? Or is it a theory that you are testing? Is there a citation for this theory?

Our statement is a little misleading and has been rephrased to express that because ice strength in the model is a linear function of the ice thickness, see equation 4 on page 5, or (Coon et al., 1974; Hibler, 1979), and the fracture angle depends on ice strength, it implicitly also depends on the ice thickness.

We included on page 13 of the revised manuscript the sentence :*Note that the ice strength is linearly related to the ice thickness (Eq. 4). Therefore the normal stress at the edge of the floe is completely defined by the thickness of the surrounding ice.*

**R1#41, Page 13 Line 17** You start this paragraph with statements about the link between initialised faults and deformation. Is this a theory you are testing? If so give a citation. Is it your interpretation of the results? if so you need to state the results and the reference the figure first.

This is a hypothesis that we can support by our simulations shown in Fig.9. The hypothesis was formulated previously and also tested (Aksenov and Hibler, 2001) and (Hibler and Schulson, 2000). It has not been tested using models whether the angle of fracture is dictated by in-homogeneities in the sea ice cover or the yield curve and flow rule (see appendix B1 and B2 for the elliptical and Coulombic yield curve in the VP rheology).

We rewrote the text of this section on page 14 of the revised manuscript : *"So far, all initial conditions have been homogeneous in thickness and concentration within the ice floe. In practice, sea ice (in a numerical model, but also in reality) is not homogeneous. A local weakness in the initial ice field is likely the starting point of a crack within the ice field (e.g., Herman, 2016, her Figure 5c). Local failures raise the stress level in adjacent grid cells and a crack can propagate. Note that the crack propagation in an "ideal" plastic model such as the VP model is instantaneous and this propagation is not seen between time steps. As a consequence, lines of failure will likely develop between local weaknesses. The location of weaknesses in the ice field together with the ice rheology (yield curve and flow rule) both determine the fracture angles (Hibler and Schulson, 2000; Aksenov and Hibler, 2001).*

*To illustrate this behavior, we start new simulations from an initial ice field with two areas of zero ice thickness and zero ice concentration, hence*

*weaker ice (Figure 9a). After $5\,s$ these simulations yield fracture patterns that are dramatically different from those of the control run simulation (Section 3.1): the fracture lines now start and terminate at the locations of the weak ice areas. Still, changing the shear strength of the ice (by changing $e$) changes the fracture pattern (Figure 9b and c). With $e = 1$, the angles are much wider than with $e = 2$, which is consistent with the general dependence of fracture angles on $e$ (see Sect. 3.3.1). Our simulations cannot lead to conclusive statements about the relative importance of heterogeneity of initial conditions and yield curve parameters for the fracture pattern, but we can state that both affect the simulations in a way that requires treating them separately to avoid confounding effects. Details are deferred to a dedicated study."*

**R1#42, Page 14 Figure 8** caption is lacking detail. Is this the figure for the lateral confinement experiments, or ice thickness experiments? Please describe what is shown in every pane.

All captions have been re-written. This is the lateral confinement experiment. Thanks for pointing this out.

The caption of figure 8 have been modified to be : *"Maximum shear strain rates (left) and stress state in stress invariant space (right) after 5 seconds of integration for different confinement pressure: $h_c = 0.05\,m$ (a) and $h_c = 0.3\,m$ (b). Note, how stress states with divergent strain rates (a) migrate left towards convergent strain rates (b)."*

**R1#43, Page 15 Figure 9** please label colour scale legends.

Corrected as suggested.

**R1#44, Page 16 Figure 11**, again please describe the simulation this figure corresponds to? The colorscale are saturated so consider rescaling.

In the case of the top left panel (now panel a) ), the log-scale of the colorbar does not allow me to use a non-saturated colormap. There is lot a simulated ice that deforms really slowly (viscous creep) These areas are not interesting for us, so it is not necessary to display them in the colormap. Additionally, logarithm of values close to zero are close to $-\infty$, so impossible to display on a colormap. We use a logarithmic colormap to have a better contrast of value of deformation, that have a really steep changes. We rewrote the caption, see below.

The caption for this figure now read *"Maximum shear strain (a), ice thickness anomaly (b), divergence (c) and stress state in stress invariant space (d) after 5 sec of integration for a smaller ellipse aspect ration ($e = 0.7$ compared to $e = 2$ in the reference run in Sect. 3.1). Compared to the control run on Fig. 3, the angle of fracture is larger ($\theta = (61 \pm 1)°$), the stress states are in the second half of the ellipse (with strain rates pointing into the convergent direction) and there is convergence along the fracture lines (panel b) in agreement with the schematic in Fig. 4"*

**R1#45, Page 17 line 18** A lot of LKFs - how many? Or is it more than compared to another simulation? rephrase or quantify.

We do not expect uniform piece of modelled ice to break in any other way than with 2 fractures. We rephrased this sentence to explain why the creation of more than 2 conjugate faults is problematic

We rephrased and changed the sentence by *Sea ice shear strength is small*

*for small stresses, and ice deforms strongly along the ice edge. Many small LKFs develop, but no large fractures spanning the entire floe, as expected in a uni-axial compressive test with an homogeneous plastic material.*

**R1#46, Page 17 line 21** space before theta
Corrected as suggested.

**R1#47, Page 18 Figure 12** are these Coulombic curve simulations? again this caption needs more detail.
Corrected as suggested
The captions of all figure have been extended to be self-sufficient and the suitable sections referenced.

We changed to caption to *"Maximum shear strain (top) and stress state in stress invariant space (bottom) for different internal angles of friction. (a) $\mu = 0.7$ or $\phi = 44°$, (b) $\mu = 0.85$ or $\phi = 58°$ and (c) $\mu = 0.95$ or $\phi = 72°$ after $5\,s$ of integration. The angles of fracture are $\theta = 23°$, $(28 \pm 2)°$ and $41°$. Fig. 10 illustrates how $\theta$ depends on $\mu$ for a Coulombic yield curve."*

**R1#48, Page 18 line 4** 'realistic manner' how are they realistic? what simulations are you comparing to what observations? A figure reference and a citation are both needed for this statement.
We mean "look realistic" when compared with observations from RGPS and reported in Hutter et al. (2018b) or lab experiments reported in Schulson (2004). We wanted to express the way the model produces small floes that appear realistic, but may be not so. Still, we can compare to small lab experiment from Schulson and Duval (2009) and see several similarities, although at different scales.
We changed the sentence to: *The fracturing of the ice floe creates smaller floes in a realistic manner, for example, compared to to Landsat-7 images (Schulson, 2004, Figure 2)*

**R1#49, Page 19 line 12** please refer to the section or figure or both that show the resolution and scale non-dependance.
The section and figure is now referred to on page 20 of the revised manuscript.

**R1#50, Page 19 line 15** 'appear' - 'appears'
Corrected, as suggested

**R1#51, Page 19 line 26** please give a citation for the statement on granular material.
The citation Balendran and Nemat-Nasser (1993) was added

**R1#52, Page 20 line 22** Citation required for 'Arctic-wide simulation'
We added a citation to Hutter et al. (2018b)

**R1#53, Page 20 line 28** why is it unsurprising? Do mean that your results fit with previous theory and results? If so can you say this and cite them?
This is not surprising because it is the role of the yield and the flow rule to determine the deformation of the solid. We modified the sentence to clarify our opinion: *"Unsurprisingly, the yield curve plays an important role in fracturing sea*

*ice in a numerical model as it governs the deformation of the ice as a function of the applied stress."*

**R1#54, Page 20 line 33** 'The ice open and create leads' - The simulated sea ice opens and creates leads
Corrected as suggested

**References**

Aksenov, Y. and Hibler, W. D. (2001). Failure Propagation Effects in an Anisotropic Sea Ice Dynamics Model. In Dempsey, J. P. and Shen, H. H., editors, *IUTAM Symposium on Scaling Laws in Ice Mechanics and Ice Dynamics*, Solid Mechanics and Its Applications, pages 363–372. Springer Netherlands.

Balendran, B. and Nemat-Nasser, S. (1993). Double sliding model for cyclic deformation of granular materials, including dilatancy effects. *Journal of the Mechanics and Physics of Solids*, 41(3):573–612.

Bouchat, A. and Tremblay, B. (2017). Using sea-ice deformation fields to constrain the mechanical strength parameters of geophysical sea ice. *Journal of Geophysical Research: Oceans*, pages n/a–n/a.

Coon, M. D., Maykut, A., G., Pritchard, R. S., Rothrock, D. A., and Thorndike, A. S. (1974). Modeling The Pack Ice as an Elastic-Plastic Material. *AIDJEX BULLETIN*, No. 24(Numerical Modeling Report):1–106.

Dempsey, J. P., Xie, Y., Adamson, R. M., and Farmer, D. M. (2012). Fracture of a ridged multi-year Arctic sea ice floe. *Cold Regions Science and Technology*, 76-77:63–68.

Heorton, H. D. B. S., Feltham, D. L., and Tsamados, M. (2018). Stress and deformation characteristics of sea ice in a high-resolution, anisotropic sea ice model. *Phil. Trans. R. Soc. A*, 376(2129):20170349.

Herman, A. (2016). Discrete-Element bonded-particle Sea Ice model DESIgn, version 1.3a - model description and implementation. *Geoscientific Model Development*, 9.

Hibler, W. D. (1977). A viscous sea ice law as a stochastic average of plasticity. *Journal of Geophysical Research*, 82(27):3932–3938.

Hibler, W. D. (1979). A Dynamic Thermodynamic Sea Ice Model. *Journal of Physical Oceanography*, 9(4):815–846.

Hibler, W. D. and Schulson, E. M. (2000). On modeling the anisotropic failure and flow of flawed sea ice. *Journal of Geophysical Research: Oceans*, 105(C7):17105–17120.

Hutchings, J. K., Heil, P., and Hibler, W. D. (2005). Modeling Linear Kinematic Features in Sea Ice. *Monthly Weather Review*, 133(12):3481–3497.

Hutter, N., Martin, L., and Dimitris, M. (2018a). Scaling Properties of Arctic Sea Ice Deformation in a High-Resolution Viscous-Plastic Sea Ice Model and in Satellite Observations. *Journal of Geophysical Research: Oceans*, 123(1):672–687.

Hutter, N., Zampieri, L., and Losch, M. (2018b). Leads and ridges in Arctic sea ice from RGPS data and a new tracking algorithm. *The Cryosphere Discussions*, pages 1–27.

Ip, C. F., Hibler, W. D., and Flato, G. M. (1991). On the effect of rheology on seasonal sea-ice simulations. *Annals of Glaciology*, 15:17–25.

Kreyscher, M., Harder, M., and Lemke, P. (1997). First results of the Sea-Ice Model Intercomparison Project (SIMIP). *Annals of Glaciology*, 25:8–11.

Lemieux, J.-F. and Tremblay, B. (2009). Numerical convergence of viscous-plastic sea ice models. *Journal of Geophysical Research: Oceans*, 114(C5).

Linow, S. and Dierking, W. (2017). Object-Based Detection of Linear Kinematic Features in Sea Ice. *Remote Sensing*, 9(5).

Marsan, D., Weiss, J., Larose, E., and Métaxian, J.-P. (2012). Sea-ice thickness measurement based on the dispersion of ice swell. *The Journal of the Acoustical Society of America*, 131(1):80–91.

Mohammadi-Aragh, M., Goessling, H. F., Losch, M., Hutter, N., and Jung, T. (2018). Predictability of Arctic sea ice on weather time scales. *Scientific reports*, 8.

Overland, J. E., McNutt, S. L., Salo, S., Groves, J., and Li, S. (1998). Arctic sea ice as a granular plastic. *Journal of geophysical research*, 103(C10):21845–21868.

Overland, J. E., Walter, B. A., Curtin, T. B., and Turet, P. (1995). Hierarchy and sea ice mechanics: A case study from the Beaufort Sea. *Journal of Geophysical Research: Oceans*, 100(C3):4559–4571.

Pritchard, R. S. (1988). Mathematical characteristics of sea ice dynamics models. *Journal of Geophysical Research: Oceans*, 93(C12):15609–15618.

Rothrock, D. A. and Thorndike, A. S. (1984). Measuring the sea ice floe size distribution. *Journal of Geophysical Research: Oceans*, 89(C4):6477–6486.

Schulson, E. M. (2004). Compressive shear faults within arctic sea ice: Fracture on scales large and small. *Journal of Geophysical Research: Oceans*, 109(C7):C07016.

Schulson, E. M. and Duval, P. (2009). *Creep and fracture of ice*, volume 432. Cambridge University Press Cambridge.

Tremblay, L.-B. and Mysak, L. A. (1997). Modeling Sea Ice as a Granular Material, Including the Dilatancy Effect. *Journal of Physical Oceanography*, 27(11):2342–2360.

Wilchinsky, A. V., Feltham, D. L., and Hopkins, M. A. (2010). Effect of shear rupture on aggregate scale formation in sea ice. *Journal of Geophysical Research: Oceans*, 115(C10):C10002.

Zhang, J. and Hibler, W. D. (1997). On an efficient numerical method for modeling sea ice dynamics. *Journal of Geophysical Research: Oceans*, 102(C4):8691–8702.

---

## Author Comment (AC2) · 14 Feb 2019

**Answer to tc-2018-192-RC2 – Jennifer Hutchings**

February 14, 2019

**Note:**

- The referees comments are shown in black.
- **The authors answers are shown in bold typeface and colored in blue.**
- **The modifications brought to the manuscript are shown in bold typeface and colored in gray.**

**Global comments**

**R2#1,** I am really pleased to see a paper that is making suggestions for idealised experiments we can use to differentiate between rheological models for sea ice. This in itself is worth publishing. The main result of the paper is that the elliptical rheology is inappropriate for representing observed cracking orientation in the ice pack, which is interesting and helps motivate changing sea ice rheological models used in climate and weather prediction.

**We thank the reviewer for the numerous interesting comments about our work. We tried below to answer and address all of them in this new manuscript.**

**R2#2,** I do have some concerns that the interpretation of observational data needs sharpening, and the results must not be overly interpreted given limitations of the use of RGPS data. Identifying intersection angles for lead pairs actually requires more work (and dedicated field data collection) than this paper warrants. You are motivated by the fact that a simulation shows larger intersection angles than RGPS, and I agree that this is something to address. I just do not think you can use RGPS to determine what the intersection angle should be, just that it needs to be smaller. Note, there are others in the community funded to do this work of identifying fracture patterns associated with particular modes of failure. For example I have an NSF and NASA project that is looking at identification of modes of failure from satellite imagery. There are also upcoming field experiments that could provide case studies to constrain the actual behaviour of sea ice, which should provide further guidance for use of your idealised cases to constrain rheological model design. I would be happy to talk to you in person about using this analysis and data to support future model validation efforts. I would also caution you to be more careful in your description of the differences between VP and granular models. Some clarification missing from the manuscript is provided in my comments. I also have suggestions for why the VP model creates LKFs, which I feel is important for understanding the validity of LKFs in the viscous plastic sea ice model representing nature.

**We agree with you that determining the intersection angle of conjugate faults from the RGPS LKF data-set has a few limitations (only large cracks, temporal resolution**

of 3 days,...). Given the large variety of forcing conditions, the RGPS LKF data-set includes LKFs originating from multiple modes of failure, but also shows conjugate fault pairs. We, here, name two advantages of using the RGPS LKF data-set to evaluate intersection angles in Pan-Arctic sea-ice simulation: (1) the data-set covers 65% of the Arctic Ocean and spans over twelve years, which is a much higher coverage compared to hand-picked studies in satellite imagery (Erlingsson, 1988; Walter and Overland, 1993). (2) The data-set enables a consistent comparison with model output as it is based on sea-ice deformation. As the intersection angles are consistent with other studies (e.g. Walter and Overland, 1993) we are confident that this approach can be used to determine whether a model is over- or underestimating the intersection angle. We, here, want to stress that we only use the misfit of intersection angles in the RGPS LKF data-set and in a Pan-Arctic sea ice simulation to motivate our work to further study the dependency of the rheology and yield curve on the intersection angle. In our manuscript, the RGPS LKF data-set is not used to evaluate our idealized experiments. We rewrote the corresponding abstract accordingly to make this point clear.

**Specific comments:**

**R2#3,** Check spelling throughout the manuscript. Also check for missing brackets throughout. Grammar can be improved in places, and sometimes words are repeated. Make sure you have someone very carefully proof read the manuscript. I did not correct all the typos I saw because I am short on time and wanted to focus my attention on the central messages in your paper.

We have carefully proofread the manuscript. We apologize for the many technical problems.

**R2#4,** In the introduction be specific that you are considering conjugate fault pairs that form under specific confining stresses, the orientation of which is controlled by the yield curve shape and flow rule. In particular reference Pritchard in the introduction. It is only when I got to the conclusion that I saw you were aware of this work and it was motivating your study. It is wise to point out that not all applied stress will result in intersecting fault pairs (for example tension and pure compression do not).

We included a citation of Pritchard (1988), and clarified that pure compression and tensile cracks do not form pair of intersecting fault. Thanks for pointing this out.

We added this text in the introduction : *"Pritchard (1988) investigated the yield curve's mathematical characteristics and derived angles between the principal stress directions and characteristics directions that depend on the tangent to the yield curve. These results show that stress states exist in plastic materials where no LKFs form and were later used to build a yield curve (Wang, 2007)."*. We also changed the first sentence of the last paragraph of the introduction by : *In this paper, we simulate the creation of a pair of conjugate faults in an ice floe with two different VP rheologies in an idealized experiment at an unprecedented resolution of 25 m. We explore the influence of various parameters of the rheologies and the model geometry (Scale, resolution, confinement, boundary conditions, and heterogeneous initial conditions).*

**R2#5, Page 2 line 4.** The appropriate references for efficient solution is Hutchings et al. 2004 or Jean-Francois Lemieux et al. 2010, I would not call LSOR or Hibler's method, which I used in the 2005 paper, as efficient. This introduces the efficient solution method that correctly couples P and U, for a convergent plastic solution. This solution

method was not used in Hutchings et al. 2005. Hutchings et al. 2005 is the correct reference for qualitatively reproducing LKFs in the viscous plastic model.

Thank you, we corrected the citations.

**R2#6, Page 2 line 15:** MEB? typo? I think you need to introduce the acronym for the Maxwell-Elastic-Brittle model here.

Corrected as suggested. We replaced "*Viscous*" by "*Maxwell (viscous)*"

**R2#7, Page 2, line 24:** argues → argued.

Corrected as suggested.

**R2#8, Page 2, line 31:** Flato and Hibler 1992 is not a mohr colomb relationship. The cavitating fluid behaves very differently and the first SIMIP (see work by Kreyscher and Harder) indicated this was not a suitable stress-strain relationship for sea ice. Also check that Ip et al. 1991 is not using a different flow rule to Tremblay's. I am wondering if you are missing text here, as these two references were left hanging

The mohr-coulomb yield curve was presented in the appendix of the Flato and Hibler (1992) paper as a possible extension to the cavitating fluid sea ice model. The way we referenced it in the paper was mis-leading.

We replaced the last sentences of this paragraph by : " *Alternative VP rheologies were never widely used in the community. These include a Coulombic yield curve with a normal flow rule (Hibler and Schulson, 2000), a parabolic lens and a tear-drop (Pritchard, 1975), a diamond-shape yield curve with normal flow rules (Zhang and Rothrock, 2005), a Mohr-Coulomb yield curve with a double-sliding deformation law (Tremblay and Mysak, 1997) or a curved diamond (Wang, 2007).*"

**Page 2**, some important points that I do not think are clear in your introduction:

**R2#9,** The Elastic-Plastic model developed during AIDJEX was based on assumptions of a material with embedded cracks in all directions that are sub-grid scale. This is closer to a ductile material than granular material.

Assuming that cracks are present in the pack ice in all direction was used to justify the isotropic assumption in the Coon et al. (1974) - later corrected in Coon et al. (2007) where the authors argued that an anisotropic assumption should be used instead. The coarse resolution of sea ice models did nothing to motivate taking into account the granular nature of sea ice in early works on sea ice models.

We have clarified this point in the revised introduction on page 2 of the revised manuscript as "*Originally, Coon et al. (1974) assumed sea ice to have cracks in all directions, justifying isotropic ice properties and isotropic rheologies.*"

**R2#10,** The Viscous-Plastic model is only considered valid on coarse resolution (Hibler 1977). It is possible to consider this model with the ice always being in a state of plastic failure, until you get to high resolutions that allow representation of ice areas between fractures, when the viscous creep, while numerically small, is unphysical. At small scales an elastic model is appropriate for low stress states. The viscous behaviour inside the yield curve is often treated as regularisation required for numerical solution.

It is true that VP rheologies are valid only at coarse resolution, but a lot of recent works feature the use of high-resolution simulation with VP models that already break this assumption (e.g. Wang et al., 2006; Hutter et al., 2018) We also think that Viscous

behavior is a regularisation of small deformations for the numerics. However it looks to us like a detail that may not need to included in the introduction. We propose to add this in the description of the VP rheology in section 2.2

We added in page 2 of the revised manuscript: *"At any scale, the assumption of viscous creep for small deformations is not physical and an elastic model would be appropriate for low stress states. The long viscous time scale, compared to the synoptic time scale of LKFs, of order 30 years (Hibler, 1979), however, allows viscous deformation to be viewed as a small numerical regularization with little implications for the dissipation of mechanical energy from the wind or ocean current (Bouchat and Tremblay, 2014), and the ice model can be considered as an ideal plastic material."*
We also added on page 6 of the revised manuscript the sentence *"Internal ice stress below these thresholds leads to highly viscous (creep) flow that parameterizes the bulk effect of many small reversible elastic deformation events. The timescale of viscous deformation is so high ($\simeq 30$ years) that viscous deformation can be seen as regularisation for better numerical convergence in the case of small deformation.."*

**R2#11,** Personally I think it is still not clear that the failure mode of a single floe is the same as an aggregate of floes. This has not been shown observationally or with models, and statements of scale invariance based on observed qualitative correspondence between failure modes in the lab (cm scale) and ice pack (10-100km scale) do not extend to the floe scale.

By conception, sea ice models used today are scale independent and are being used at resolution approaching the floe scale. We added a sentence to specify that the fracture process at the floe scale as not been shown to have the same failure mode as at arctic and lab scales.

We added a sentence on page 3 of the revised manuscript: *"The scale invariance of the fracture processes at the floe scale has not yet been shown, especially due to the lack of observations at both high spatial and temporal resolution."*
We also added on page 2 of the revised manuscript: *"It can be argued that if the mode of deformation of a single floe is similar to that of an aggregate of floes, a given rheology developed for a continuum can still be applicable at spatial resolutions of the order of the floe size (Overland et al., 1998), but the validity of a given flow rule across scales is not clear."*

**R2#12, Page 3** Discussion regarding orientation of intersecting LKFs from RGPS: I performed a similar analysis back in the early 2000's and never published the result, which was a wide spread in intersection angle. The reason I did not publish this is because I realised that the RGPS product could potentially be capturing fracture zones that form at different times, and therefore in the product appear to be a conjugate pair because they intersect, but they are not because they were not formed under the same confining stress. This is really obvious if you spend some time on the ice pack in winter and observe leads forming and working. RGPS is not the right satellite product to use to identify conjugate fault pairs in sea ice. Hence I disagree that you can state "The wide range of intersection angles is presumably due to previous deformation history and associated hetrogeneity in the ice cover that dictates the strength locally".

Thanks for pointing this out here and also in a recent conversation with co-author Tremblay. It is correct that RGPS data represent a mean over 3 days and for this reason

we cannot be certain that intersecting fractures were formed simultaneously. In revising the paper, we have downplayed the RGPS as a dataset used for validation/motivation as per your suggestion and that of the other reviewer.

**R2#13, page 3, line 32:** Just want to clear up one very important point about my 2005 paper. It is steep stress gradients in the model sea ice stress field that allow LKFs to form. I suspect this opening is related to an instability in the model identified by Nico Gray (Gray and Kilworth 1995). We seeded stress gradients though a random number being added to P* (which defines compressive strength). At the time I wrote this paper I was obsessed with plastic convergence of the VP solution, so made sure there were no spurious stress values due to the numerical error. The pan-Arctic model of Heil's included in this paper, and other VP models, are able to show LFKs because of the noise introduced by not converging fully to the yield curve. If you play around with a VP model you can create divergence related instabilities along gradients in forcing (e.g. non-smoothly interpolated wind fields), or even have the model blow up and crash due to one localised discontinuity in thickness (e.g. I have seen this when using a nudging method to assimilate data into the CICE model that created open water locally). The reason I bring this up is that I feel it is very important that people understand how the VP model can create LKFs. The mechanism is quite different from what might actually be happening in a granular or brittle material. I would be very happy to advise on experimental design, repeating and following up on my investigations 15 years ago.

Simulation by Lemieux and Tremblay (2009) show LKFs in a fully converged solution using the JFNK method (Lemieux et al., 2010) using a realistic but smooth thickness and concentration field. In the response to reviewer document for the Lemieux and Tremblay paper (not published), we also showed LKFs in idealized experiments with a constant thickness and concentration field. In the present paper, we also show clear discontinuity in the strain rate fields that becomes apparent in the thickness and concentration field after some integration. We see the VP model as an ideal plastic model as opposed to a viscous plastic model given that the time scale associated with the viscous term (for the default $\eta_{\max}$ an $\zeta_{\max}$) is $\sim 35$ years and LKFs form over time scale of a few days. So, for all practical purposes, the viscous term does not operate on time scale of interest to LKFs formation. Ideal plastic material in turn can be viewed as an elastic-plastic material with an infinite elastic wave speed (stresses adjusts instantaneously with the forcing in in the "elastic" regime and can form linear kinematic features. For all these reasons, we think that the instability described in Gray and Killworth (1995) is not responsible for the formation of LKFs in a VP model. A formal comparison between elastic-viscous-plastic (MEB) model and a viscous-plastic model is underway by one of the co-authors. This will be studied in more details in that paper. Further discussion with the reviewer on this topic will be very welcome.

**R2#14, page 3, line 34:** The original study on shape of yield curve and ice arches is in Billy Ip's thesis, that was published later by Hibler in Hibler et al. (2006).

This reference was added to the introduction on page 3 of the revised manuscript. Thanks for pointing this out.

**R2#15, page 8, line 14.** A reader unfamiliar with numerical solution of the VP model will need some guidance as to what non-linear and linear iterations are. I know you are talking about the sub-cycling to reach plastic equilibrium (or close to it) and converge the velocity solution at each time step. Perhaps use language that is more obvious to a casual reader. Incidentally, did you check convergence properties? Just curious. I think you point out somewhere that the modified coulombic rheology is slower to converge -

I found that solutions for yield curves with corners never converged fully. A frustrating reality! If you follow my suggestions to delve into why the model creates LKFs you will need a full description of the interative process and convergence characteristics.

Our theory of yield curve added in appendix B gives an explanation of why a yield curve with corners gives poor convergence. We modified this paragraph to improve clarity about the LSR solver scheme and the presence of sub-cycles (or outer-loops), as also asked by the other referee.

We modified the text describing the numerical solver : *We solve the non-linear sea-ice momentum equations with a Picard or fixed point iteration with 1500 non-linear or outer-loop (OL) iterations. Within each non-linear iteration, the non-linear coefficients (drag coefficients and viscosities) are updated and a linearized system of equations is solved with a Line Successive (over-)Relaxation (LSR) (Zhang and Hibler, 1997). The linear iteration is stopped when the maximum increment is less than $\epsilon_{LSR} = 10^{-11}\, m\, s^{-1}$, but we also limit the number iterations to 1500. Typically, 1500 non-linear iterations are required to reach a converged solution. This is so because of slow convergence due to the highly non-linear rheology term and the high spatial resolution (Lemieux and Tremblay, 2009).*

**R2#16, page 8**, results section: Describe what the applied strains are in the numerical experiments (magnitude, not just direction).

The specified strains are described in equation 17 on page 8 of the original manuscript (or page 8 of the revised paper), and their magnitude is documented in Table 1. We use the same strains for all experiments, excepted for the up-scaled experiment where the magnitude of the strain is up-scaled as well.

**R2#17, Page 8 line 26:** What are the default parameters? I think you forgot to reference table 1.

We added the reference to Table 1 on page 9.

**R2#18, page 9, line 4:** Regarding your statement "Fracture occurs when the stress state intersects the yield curve". Plastic failure occurs then. The fact that a "fractures" form is because the ice deforms at a stress discontinuity where the stress accumulates and reaches yield. You are correct in pointing out that the strain-rate has characteristic directions along which divergence will occur, defined by the shape of the yield curve and flow rule. It is this divergence, relative to the confining stress, that defines the directions of the linear deformation features in the model runs.

We agree on this, this paragraph have been modified to clarify this point.

This sentence has been rephrased as *Fracture occurs after plastic failure when the stress state reaches the yield curve and the ice starts to move in divergence.* for clarity.

**R2#19, Comment on differences between 3.1 and 3.2:** The change in nature of cracks when you decrease the number of internal iterations (linear iterations) is probably related to the fact that the ice stress field is more heterogeneous (further from the converged solution) and the LSOR method tends to create noise in the stress field and smoothes with increasing number of iterations (unlike the SIMON method I proposed, Hutchings et al. 2004, that has a smoother convergence to the yield curve). Hence there are more points where LKFs can nucleate when you reduce the number of internal iterations. This is just a suggestion, with out looking at the stress fields in your experiments I can not tell you if this is what is actually happening. Incidentally, another unpublished result that I presented at AGU in 2003: The VP model can create intersecting deformation

features across the entire Arctic Ocean is you do not converge to plastic equilibrium and are not careful in smoothing the solution between time steps (which can be done numerically through the choice of advection scheme or Bill's introduction of artificial diffusion in his 1979 paper). I never followed up this work. I suspect that this is a direct consequence of Gray's instability. This instability is damped by the addition of numerical diffusion (or artificial diffusion) in the solution proceedure. We might think the resultant strain-rate fields are more realistic, I just do not believe using the non-convergence and numerical instability is an appropriate way to model the process because we are not controlling the nature of the stress concentrators or stress propagation in the model appropriately. The key point is that the ice pack strength is highly heterogeneous and while we do not know the nature of the stress concentrators in the ice pack, they are likely to be more randomly distributed (which non-convergence to the yield curve might be approximating, but is not controlled for). There is a need to understand the nature and distribution of the stress concentrators in sea ice, so we can appropriately model this. And it would really help future sea ice modellers to point out this issue more clearly in papers that investigate LKFs in the viscous-plastic model.

We are afraid that there is a misunderstanding. The results of section 3.2 (3.2.1 in the revised manuscript) do not critically depend on the number of non-linear iterations. The section 3.2 uses less iterations because we wanted to run the idealized experiment for a longer time, but we do not intend to compare the effect of the number of iterations. If we use the same low number of iterations for the experiments in section 3.1, we obtain almost the same shear pattern although the shear lines are not so well defined. We improved the description in the beginning of 3.2 to try to avoid any misunderstandings. We more than agree with the last statement in this comment.

The description of this experiment is now in Section 3.2.1 of the revised manuscript on page 12 and reads "*Continuing the integration to 2700 seconds (45 min), compared to 20 seconds in the reference simulation leads to the creation of smaller diamond-shaped ice floes due to secondary and tertiary fracture lines (Figure 5). The openings are visible in the thickness and concentration fields with thinner, less concentrated ice in the lead. In this longer experiment, the sea ice also ridges, for instance at the center of the domain where the apex of the diamonds fails in compression. There is also some thicker ice at the northern boundary induced by the specified strain rate at the northern boundary. The fracture pattern and presence of secondary and tertiary fracture lines are in line with results from laboratory experiments Schulson (2004) and with AVHRR and RGPS observations.* ".Figure 6 is on page 14 of the revised manuscript*

**R2#20, Figure 7:** Nice illustration of the role of boundary conditions on the stress solution.

We thank the reviewer.

**R2#21, Section 3.5:** Good illustration. I would suggest you critically look at the stress fields in your previous experiments to identify what the stress concentrators are there. Did you forget to reference figure 9 in this section. Finally, Bill Hibler has shown similar results where embedded fractures of different orientations would join together to form larger scale fracture patterns. Not sure he published that, but I think he did. Look at the papers he wrote with Aksenov and his first anisotropic paper with embedded leads in grid cells. Unfortunately I am on an airplane right now and don't have access to his papers.

A reference to figure 9 is present on line 17 of the original manuscript. We added the reference to Aksenov and Hibler (2001) In the revised manuscript on page 15. Thanks for pointing this out. We do not have stress concentrators in the previous experiments, except for Figure 7b where the no slip southern boundary forces the fracture to take angle solely determined by geometry.

**R2#22, page 16 line 1:** Here and in other places you confuse the simulation with reality. "This is in contrast with other granular materials". Remove "other", as this is in contrast with granular materials. The VP model is not modelling a granular material. While sea ice may be a granular material, the rheology is designed for different behaviour.

"Corrected as suggested. Thanks for picking this up. We do understand that model and reality are different!.

**R2#23, page 16 line 5:** "larger that what" → "larger than that"
Corrected as suggested.

**R2#24, page 16, line 9:** I would like to see the strain-rate field for the longer simulation with e=0.7 where deformation is in convergence. Where in the field is the ridging occurring? What do the intersection angles look like?

Figure 11 shows the fracture for $e = 0.7$ after 5 seconds. We added a figure showing the ice field with e=0.7 for 2700 seconds (45 minutes) o, similar to Figure 6.

**R2#25, page 17 line 2:** "and individual floes form" could be clarified as "4 separate floes form". Corrected as suggested.

**R2#26, page 17 line 15:** Please clarify the statement "the fracture pattern is very sensitive to coefficient of internal friction. This makes measuring the fracture angle very difficult." Surely the sensitivity will help you differentiate fracture angles. Incidentally mu was not defined in section 2 or here. What causes the spread in the stress state? Is this related to the opening/ridging and subsequent ice strength changes? So the spread in stress state is controlled by the strength parameterisation. I feel this is important to point out, because it is another control we have on the spread of intersection angles you might see under a particular confining stress.

We wanted to express that we observe different behavior depending on the value of $\mu$. We revised the text by rewording this part along a whole paragraph on page 20 of the revised manuscript.

we also added a sentence in section 2.2 defining $\mu$ : "[..] where $\mu$ is the slope of the Mohr-Coulomb limbs (Fig. 1), $c$ is the cohesion value (the value of $\sigma_{II}$ for $\sigma_I = 0$) defined relative to the tensile strength by $c = \mu \cdot T^{\star}$."

**R2#27, page 17 line 25:** Clumsy language: "the stress state touches the yield curve on both parts of the yield curve." Very unclear that you mean the stress state falls on the coulombic limb and the ellipse cap. rephrase.

We modified the whole section describing the Coulombic yield curve experiments, on page 18-19 of the revised manuscript. We hope it clearer now.

**R2#28, page 19 top paragraph:** This is a matter of opinion. I disagree that the ice pack is characterised by diamond shaped floes. Yes, diamond floes form under certain confining stresses, but is this the most prevalent mode of failure in winter? That needs to be proven. This point does not discount your use of your numerical experiments to differentiate between rheologies, but it does question if an anisotropic rheology based on diamond shaped floes is appropriate for all space and time.

[Figure]

Figure 1: Sea ice thickness (a), concentration (b), maximum shear strain rate (c) and divergence (d) after 45 min of integration (2700 sec) in a uni-axial loading test with an ellipse ratio $e = 0.7$. To make these longer simulations possible, both non-linear and linear iterations are limited to 150 per timestep. Results show that no fracture lines are created, but the ice is pilling close to the northern boundary and the ice got broader without creating open-water.

We clarified this point in the revised manuscript on page 21 of the revised manuscript. We modified the text on page **20-21** of the revised manuscript to state : *"The Elastic Anisotropic Plastic (EAP) rheology assumes predominately diamond shaped floes in sea ice (Wilchinsky and Feltham, 2006). A sea ice model with EAP creates sharper fractures than a model with the Elastic Viscous Plastic (EVP, Hunke and Dukowicz, 1997) rheology (Heorton et al., 2018). The authors concluded that the anisotropic model may improve the fracturing process for sea ice, especially by creating areas of oriented weaknesses, and particularly at coarse resolution where the fracture is not resolved by the grid spacing. In the experiments presented here, the VP rheologies lead to sharp and anisotropic fracture lines without any additional assumptions."*

**R2#29,** Also, did you calculate characteristic directions for the VP model to confirm these are controlling the diamond structures in your simulations? I have a code somewhere (from 15 years ago) that does this. If I can find it I can give it to you.

No we did not. But using the theory we described for the Coulombic yield curve in the original manuscript, we can know predict the fracture angle for the elliptical yield curve. This theory is presented in Appendix B of the revised manuscript.

**R2#30, page 19 line 12:** This sentence is miss-representative: "Thus, the rheology is shown to be scale independent ... in line with observations". Your numerical experiment is set up to ensure the behaviour is scale independent from the scale of the grid size to the domain size. You would really hope your numerical experiment results do not depend on resolution (good practice to check this) and there is no reason scale should change intersection angles for the reasons you have stated previously. Rephrase this statement so someone does not quote it as evidence supporting Schulson's hypothesis.

We modified the sentence on page 21 of the new manuscript to be *"The fracture angles do not depend on the spatial resolution and domain size as expected in our idealized numerical experiment setup (Sect. 3.2.1, Fig. 5)"*

**R2#31, page 19, line 20:** Just to clarify, the reason the experiments with thin ice change the fracture angle is because the presence of the thin ice modifies the stress state across the domain. So with an ellipse this will change the intersection angle, with a coulombic rheology it would not. I feel this part of the paragraph needs more clarification.

we modified the sentence on page 21 of the revised manuscript :

*"The confining pressure (i.e. thin ice imposed on the side of the domain) changes the distribution of stress within the domain. This results in different deformation patterns (shear and divergence) and different fracture angles because the yield curve is convex and uses a normal flow rule."*

**R2#32, page 19 line 22:** Your interpretation of the RGPS data (if one believes the intersection angles are at conjugate pairs and not leads formed at separate times) would lead one to believe that there is not a constant fracture angle independent of confining stress.

The macroscopic angle of friction in a granular material is not constant and depends on the distribution of the contact normal between floes (Balendran and Nemat-Nasser, 1993). A consequence of a variable macroscopic angle of friction is fracture angle that is not constant. This work is beyond the scope of the present paper.

**R2#33, page 20 line 2:** Perhaps clarify that the Miller et al. (2005) experiments were using metrics of ice thickness, area and velocity to determine the optimal yield curve shape. It is my memory they did not consider the form of the ice strength parameterisation as an alternative to changing shear strength, or yield curve shape, just the eccentricity of the ellipse.

We think that the metrics do not matter in this context. We would like to keep it simple here, because we are only referring to change of $e$.

We added on page 21 of the revised manuscript *"Arctic-wide simulations improve metrics of sea ice concentration, thickness and velocity by decreasing the value of $e$ of the standard elliptical yield curve, that is, by adding shear and bi-axial tensile and compressive strength (Miller et al., 2005; Ungermann et al., 2017)."*

**R2#34, page 20 line 5:** Can you show that your numerical experiments are consistent with Pritchard (1988).

Yes, we can. The fracture angle relative to principal stress corresponds to the angle between the principal stress and characteristics directions given by Pritchard (1988) or Wang (2007). We have added an appendix B with a theory explaining the fracture angle of the yield curve using Mohr's circle. Theory that gives the same relation between the slope of the tangent to the yield curve and the fracture angle.

**R2#35, page 20 line 8:** questions → questioned

Corrected as suggested.

**R2#36,** After reading your discussion I wondered if the tear drop yield curve (originally proposed by Pritchard) might be more appropriate that the Hibler modified ellipse / mohr coulomb.

We note that the kink in the MC yield curve of Hibler and Schulson (2000) cannot be eliminated by choosing appropriate values of P* and e. Also, both tear drop and ellipical yield curve use the normal flow rule and have a convex yield curve, which gives the non-physical behavior of the fracture angle as a function of shear strength and confining pressure. It is also known that a normal flow rule with "straight limbs" gives too much dilatation when the stress states are lying on the straight limbs (as stated in Flato and Hibler, 1992). For this reason we believe that a Mohr-Coulomb or a tear drop with a non-associated (normal) flow rule (e.g. Tremblay and Mysak, 1997) would be more appropriate. This is the subject of future work.

**R2#37,** Also, Hibler and others recognise that you must have a closed cap on a coulombic rheology to allow ridging. Perhaps the ellipse is not the best choice for this. In engineering it is more common to have a flatter closure to the yield curve.

The flatter closure actually would lead to other problems, if we look at the framework we develop in the new version of the appendix ??, and also with the theory of the characteristics of the yield in Pritchard (1988). A slope of yield curve higher than $|b'| = 1$ does not have solution for fracture.

**R2#38, page 20 line 26:** sensible → sensitive

Corrected as suggested.

**R2#39, page 20 line 26:** Another example where it is not so clear you are talking about the VP rheology problem: "The fracture angles is also sensitive to the surrounding sea ice cover, in contradiction to the granular nature of sea ice". Also, the stress field is going to depend on surrounding ice even when the ice is modelled as granular, which I am not sure is what you were meaning to imply is not true for granular materials. I think you need to clarify the language, and I think I disagree with you that this test is suggesting ice is granular - it would be something we could test in an ice tank to find out what the actual behaviour is though. I feel you do not highlight a key result in the paper: That fracture angles below 30o are not possible with the elliptical rheology, and that this is in direct conflict with observational evidence for smaller fracture angles. Even in light of errors of interpretation of the RGPS intersection angles this result still holds.

Thanks for pointing this out. We have made this result (fracture angle below 30deg not possible with ellipse) more prominent on page 22 of the revised manuscript. If we think of the nature of sea ice, it is a granular material composed of floes. At no or small confinement, the ice dynamic is mainly governed by the floes lateral interaction. The difference with "classic" granular material (sand, clay,...) the ice is a 2D material bounded to float on the ocean. So at high confinement the ice can "escape" in 3D and ridge or raft. We agree that sea ice should have different behavior for high and low confinement. This is the conclusion also reached by Wang (2007) if we look at their Figure 5, even if we disagree with the shape of their yield curve.

The first paragraph of the conclusion have been replace by the 4 following :

In our experimental configuration with uni-axial compression, fracture angles below 30° are not possible in a VP-model with an elliptical yield curve. Observations suggest much lower values. We find an empirical relationship between the fracture angle and the ellipse ratio $e$ of the elliptical yield curve that can be fully explained by the convexity of the yield curve (Appendix B). In contrast to expectations, increasing the maximum shear strength in the sea ice model increases the fracture angle. Along a fracture line, there can be both divergence and convergence depending on the shear strength of the ice, linked to the flow rule. The simulated ice opens and creates leads with an ellipse ratio $e > 1$ (shear strength is smaller than compressive strength), and ridges for $e < 1$ (shear strength is larger than compressive strength).

With a modified Coulombic yield curve, the fracture angle can be decreased to values expected from observations, but the non-differentiable corner points of this yield curve lead to numerical (convergence) issues and, for some values of the coefficient of internal friction $\mu$, to fracture patterns that are difficult to interpret. At these corner points, two different slopes meet and give two non-unique solutions for fracture angles and deformation directions. We recommend to avoid non-differentiable yield curves (with a normal flow rule) in viscous-plastic sea ice models.

More generally, the model produces diamond-shaped fracture patterns. Later the ice floe disintegrates into several smaller floes develop. The fracturing process in the ice floe in our configuration is independent of the experiment resolution and scale, but sensitive to boundary conditions (no-slip or free-slip). The fracture angle in the VP-model is also sensitive to the immediate environment. This is not consistent with the notion of sea ice as a granular material. Unsurprisingly, the yield curve plays an important role in fracturing sea ice in a numerical model as it governs the deformation of the ice as a function of the applied stress.

**R2#40, page 21 line 3:** Note that at cusps in a yield curve two possible solutions are possible. I feel you can clarify your point about not using non-differentiable yield curves. They are also numerically unstable. The unclear fracture pattern is not something I have issue with. Perhaps this exists in reality when the stress state can spread across opening and closing modes.

Unclear (or chaotic) fracture pattern could happen in reality when there is an high heterogeneity in ice strength, concentration and thickness, but it shall not happen in a uni-axial compression experiment with uniform ice field.

We added the following sentence on page 22 of the manuscript : *"At these corner points, two different slopes meet and give two non-unique solutions for fracture angles and deformation directions."*

**R2#41, page 21 line 10:** I feel you can be stronger here in stating that the ellipse with normal flow rule can be discounted as unphysical.

We added a sentence in the revised manuscript:

We modified the text on page 22 of the revised manuscript *"In our experimental configuration with uni-axial compression, fracture angles below 30°*

*are not possible in a VP-model with an elliptical yield curve. Observations suggest much lower values. We find an empirical relationship between the fracture angle and the ellipse ratio e of the elliptical yield curve that can be fully explained by the convexity of the yield curve (Appendix B)."*

**R2#42, page 21 line 11:**
Scale is really unimportant in these experiments. You can perform them on any scale. The more important question is what scale do these types of fracture events actually occur on and can that be resolved in models?

As stated in comment R2#30, we agree on the fact that the scale of such idealized experiment is not important. It appeared important to us to show this fact with simulations. The standard VP model is used by the community at various scales and resolution depending on the goal of each particular studies, i.e. paleoclimate studies and sea ice prediction for ships operations, so the fact that the VP rheologies are scale-independent is important to point out. The observations of spatial power-law scaling in sea-ice deformation down shows that the fracture does not have a preferred scale. The scaling behaviour is seen at lengths ranging from basin scale in satellite observations (Marsan et al., 2004) down to 50m in ship radar observations (Oikkonen et al., 2017). At high resolution VP-simulations are able to reproduce this spatial scaling behaviour while underestimating the intermittency in temporal scaling (Hutter et al., 2018). The resolution used in our study is in between the one used in (Hutter et al., 2018) and the lower limit of scales where we observe power-law scaling.

*We replaced the aforementioned sentence by "If Arctic-wide sea ice simulations with a resolution of 25 m are not feasible today because of computational cost, we can still imagine small experiments to be useful for process modeling on small scales when local and high-resolution observations (e.g. wind, ice velocities) are available. For example, such process modeling studies could be used to constrain the rheology with data from the upcoming MOSAiC campaign (Dethloff et al., 2016) that will provide a full year of sea ice observations in pack ice."*

**R2#43,** I see you did not reference work by K. Wang (2007) who used lead intersection angles to try to estimate the shape of a yield curve. He also has a paper were he performed a similar study to you (Wang and Wang 2010), however for the pan-arctic and perhaps with convergence issues that make his findings hard to interprete. While this work suffered from problems of representativeness of the observational data (how can you be sure fractures formed at the same time), as you do, I feel you should consider Wang's papers in light of your findings.

We now included a discussion of the results and findings of these the first references On page 21 of the revised manuscript. We did not wish to include the second one because of the strong convergence issues.

*we added the text "Based on the results of Pritchard (1988), Wang (2007) used observed fracture patterns to design a* Curved Diamond *yield curve. But this yield curve also contains a non-differentiable point, which will be problematic for numerical reasons."*

**R2#44,** Finally, I believe that the stress state between fractures in your numerical experiments is inside the yield curve (viscous), and the motion close to zero. Is this correct, it was what I found when I was working on this. Just a point to clarify that the accumulation of stress along fractures is due to the yield curve discontinuity, and

the associated characteristic directions in the strain field that control the propagation of fracture direction. This accumulation of stress needs to be nucleated at a location with high stress gradient (such as a corner on the boundary or strength/stress difference between grid cells). Once the stress reaches the yield curve, the numerical instability is probably put into play during the inner iterations. You do not see LKFs in VP models that have smooth boundaries and strength fields. The formation of LKFs is grid resolution dependent (as the linear instability identified by Gray is). You have speculated on why LKFs form in the VP model only at higher resolutions in a previous paper and I would suggest the place to look is in the convergence of the solver, and the splitting of velocity solution from the ice strength (pressure). I do not think it is just the fact that divergence (and strength reduction) can be greater at higher resolution. Clarifying this mechanism will help readers understand why VP models show this behaviour. It will also hopefully get people thinking about how to represent stress accumulators in the model, because many people using the VP model and studying fractures are unaware of how the model produces these.

Jenny

The stress states outside of the LKFs are effectively inside the yield curve, at the exception of the cells on the border of the ice floe, that move into open-water. We think that the creation of fractures in the model VP is explained by the Mohr's circle and failure envelope theory and we create LKFs in our uniform ice strength field. We have added a new appendix discussing this, Appendix B.

**References cited in the referee comments**

Gray and Kilworth (1995) Stability of the viscous plastic sea ice rheology. J. Phys. Ocean. 25(5), 971-978.

J-F. Lemieux, B. Tremblay, J. Sedlacek, P. Tupper, S. Thomas, D. Huard, J-P. Auclair. 2010. Improving the numerical convergence of viscous-plastic sea ice models with the Jacobian-free Newton-Krylov method. J. Comp. Phys., 2840-2852.

Hutchings J.K., H. Jasak and S. W. Laxon 2004. A strength implicit correction scheme for the viscous-plastic sea ice model. Ocean Modelling.

Hibler W.D. III, J.K. Hutchings, and C. F. Ip, 2006. Sea-ice arching and multiple flow state of Arctic pack ice. Ann. Glaciol. 44.

Hibler W.D. III. 1977. A viscous sea ice law as a stocahastic average of plasticity.

Wang, K. 2007, Observing the yield curve of compacted pack ice, J. Geophys. Res. 112, C05015.

Wang, K. and Wang, C. 2010, Modelling linear kinematic features in pack ice, J. Geophys.Res. 114, C12

**References**

Aksenov, Y. and Hibler, W. D. (2001). Failure Propagation Effects in an Anisotropic Sea Ice Dynamics Model. In Dempsey, J. P. and Shen, H. H., editors, *IUTAM Symposium on Scaling Laws in Ice Mechanics and Ice Dynamics*, Solid Mechanics and Its Applications, pages 363–372. Springer Netherlands.

Balendran, B. and Nemat-Nasser, S. (1993). Double sliding model for cyclic deformation of granular materials, including dilatancy effects. *Journal of the Mechanics and Physics of Solids*, 41(3):573–612.

Bouchat, A. and Tremblay, B. (2014). Energy dissipation in viscous-plastic sea-ice models. *Journal of Geophysical Research: Oceans*, 119(2):976–994.

Coon, M., Kwok, R., Levy, G., Pruis, M., Schreyer, H., and Sulsky, D. (2007). Arctic Ice Dynamics Joint Experiment (AIDJEX) assumptions revisited and found inadequate. *Journal of Geophysical Research: Oceans*, 112(C11):C11S90.

Coon, M. D., Maykut, A., G., Pritchard, R. S., Rothrock, D. A., and Thorndike, A. S. (1974). Modeling The Pack Ice as an Elastic-Plastic Material. *AIDJEX BULLETIN*, No. 24(Numerical Modeling Report):1–106.

Dethloff, K., Rex, M., and Shupe, M. (2016). Multidisciplinary drifting Observatory for the Study of Arctic Climate (MOSAiC). *EGU General Assembly Conference Abstracts*, 18.

Erlingsson, B. (1988). Two-dimensional deformation patterns in sea ice. *Journal of Glaciology*, 34(118):301–308.

Flato, G. M. and Hibler, W. D. (1992). Modeling Pack Ice as a Cavitating Fluid. *Journal of Physical Oceanography*, 22(6):626–651.

Gray, J. M. N. T. and Killworth, P. D. (1995). Stability of the Viscous-Plastic Sea Ice Rheology. *Journal of Physical Oceanography*, 25(5):971–978.

Heorton, H. D. B. S., Feltham, D. L., and Tsamados, M. (2018). Stress and deformation characteristics of sea ice in a high-resolution, anisotropic sea ice model. *Phil. Trans. R. Soc. A*, 376(2129):20170349.

Hibler, W. D. (1979). A Dynamic Thermodynamic Sea Ice Model. *Journal of Physical Oceanography*, 9(4):815–846.

Hibler, W. D. and Schulson, E. M. (2000). On modeling the anisotropic failure and flow of flawed sea ice. *Journal of Geophysical Research: Oceans*, 105(C7):17105–17120.

Hunke, E. C. and Dukowicz, J. K. (1997). An Elastic–Viscous–Plastic Model for Sea Ice Dynamics. *Journal of Physical Oceanography*, 27(9):1849–1867.

Hutter, N., Martin, L., and Dimitris, M. (2018). Scaling Properties of Arctic Sea Ice Deformation in a High-Resolution Viscous-Plastic Sea Ice Model and in Satellite Observations. *Journal of Geophysical Research: Oceans*, 123(1):672–687.

Lemieux, J.-F. and Tremblay, B. (2009). Numerical convergence of viscous-plastic sea ice models. *Journal of Geophysical Research: Oceans*, 114(C5).

Lemieux, J.-F., Tremblay, B., Sedláček, J., Tupper, P., Thomas, S., Huard, D., and Auclair, J.-P. (2010). Improving the numerical convergence of viscous-plastic sea ice models with the Jacobian-free Newton–Krylov method. *Journal of Computational Physics*, 229(8):2840–2852.

Marsan, D., Stern, H., Lindsay, R., and Weiss, J. (2004). Scale Dependence and Localization of the Deformation of Arctic Sea Ice. *Physical Review Letters*, 93(17):178501.

Miller, P. A., Laxon, S. W., and Feltham, D. L. (2005). Improving the spatial distribution of modeled Arctic sea ice thickness. *Geophysical Research Letters*, 32(18).

Oikkonen, A., Haapala, J., Lensu, M., Karvonen, J., and Itkin, P. (2017). Small-scale sea ice deformation during N-ICE2015: From compact pack ice to marginal ice zone. *Journal of Geophysical Research: Oceans*, 122(6):5105–5120.

Overland, J. E., McNutt, S. L., Salo, S., Groves, J., and Li, S. (1998). Arctic sea ice as a granular plastic. *Journal of geophysical research*, 103(C10):21845–21868.

Pritchard, R. S. (1975). An Elastic-Plastic Constitutive Law for Sea Ice. *Journal of Applied Mechanics*, 42(2):379–384.

Pritchard, R. S. (1988). Mathematical characteristics of sea ice dynamics models. *Journal of Geophysical Research: Oceans*, 93(C12):15609–15618.

Schulson, E. M. (2004). Compressive shear faults within arctic sea ice: Fracture on scales large and small. *Journal of Geophysical Research: Oceans*, 109(C7):C07016.

Tremblay, L.-B. and Mysak, L. A. (1997). Modeling Sea Ice as a Granular Material, Including the Dilatancy Effect. *Journal of Physical Oceanography*, 27(11):2342–2360.

Ungermann, M., Tremblay, L. B., Martin, T., and Losch, M. (2017). Impact of the Ice Strength Formulation on the Performance of a Sea Ice Thickness Distribution Model in the Arctic. *Journal of Geophysical Research: Oceans*, pages n/a–n/a.

Walter, B. A. and Overland, J. E. (1993). The response of lead patterns in the Beaufort Sea to storm-scale wind forcing. *Annals of Glaciology*, 17:219–226.

Wang, K. (2007). Observing the yield curve of compacted pack ice. *Journal of Geophysical Research: Oceans*, 112(C5):C05015.

Wang, K., Leppäranta, M., and Kõuts, T. (2006). A study of sea ice dynamic events in a small bay. *Cold Regions Science and Technology*, 45(2):83–94.

Wilchinsky, A. V. and Feltham, D. L. (2006). Anisotropic model for granulated sea ice dynamics. *Journal of the Mechanics and Physics of Solids*, 54(6):1147–1185.

Zhang, J. and Hibler, W. D. (1997). On an efficient numerical method for modeling sea ice dynamics. *Journal of Geophysical Research: Oceans*, 102(C4):8691–8702.

Zhang, J. and Rothrock, D. A. (2005). Effect of sea ice rheology in numerical investigations of climate. *Journal of Geophysical Research: Oceans*, 110(C8):C08014.

---

## Author Response (AR2)

**tc-2018-192**
**Authors answer to editors comments**

March 20, 2019

Dear Editor,

We are really pleased with your decision to accept our manuscript pending some minor revisions. Please find below (1) our answers (in blue) to your demands (in black) and (2) the modifications we brought to the manuscript (latexdiff). After inspecting the manuscript before re-submission, we made a few minor corrections (grammar and orthography).

Yours sincerely,

On behalf of the authors,
Damien Ringeisen

**Comments to adress**

1. In the edits in response to R1#2 dealing with the continuum assumption, you cite Overland et al 1998. This point is addressed head on in Feltham [2008], Ann Rev Fluid Mech, in supplemental Appendix C. It would be appropriate to cite this work.
   **Answer:** Added as suggested.

2. In the response to R1#3 it is stated that the fracture angle provides "for the first time" a meaningful diagnostic that allows discrimination between different rheologies. I am not sure what is meant by "first time" in this context. Fracture angles and have been used to motivate sea ice rheology before [Erlingsson, 1988], and to constrain them [Wilchinsky and Feltham, J. Geophys Res, 2011; J. Phys. Oceanogr., 2012]. These latter two references probably do deserve to be cited, perhaps where you talk about initial conditions on page 14, where it is also relevant to cite/mention Wilchinsky, Feltham, and Hopkins [2011], which used a discrete element model to simulate ice fracture from an initial state of oriented cracks.
   **Answer:** The statement *"for the first time"* was referencing to the fact that we are here looking directly at the fracture and not at statistics or scaling laws. We find that the proposed citations are pertinent for creation of LKFs with heterogeneous leads.
   The following sentence has been added on page 15 of the revised manuscript: *"The influence of previous leads on subsequent lead creation have been studied with a Discrete-Element-Model (Wilchinsky et al., 2011) and has been used to constrain new anisotropic rheologies that include the effects of embedded anisotropic leads (Wilchinsky and Feltham, 2011, 2012)."*

3. In response to R1#48, I agree with your point that the model may produce small floes that appear realistic but the manner of producing them may not be realistic. However

your edit to the manuscript does not capture this point, where you say "creates smaller floes in a realistic manner". I hope/assume this is just a subtlety of English that was not appreciated. A phrasing that more accurately captures your response to the reviewer is "creates smaller floes in a manner that appears realistic".

**Answer:** Changed as suggested

4. In the edit in response to R2#11 you have stated "The scale invariance of the fracture process at the floe scale has not yet been shown, especially due to the lack of observations at both high spatial and temporal resolution." The clear implication of this is that (1) you believe such scale invariance is true; and (2) the only reason it has not been shown is because of lack of high resolution observations. Since neither you nor I have divine knowledge, and there are plenty of factors that the model, mathematical or conceptual, does not account for in relation to real sea ice, you should reword to a more modest, but defensible, statement. I suggest: "The scale invariance of the fracture process at the floe scale has not been shown. There is a lack of observations at both high spatial and temporal resolution that might show this."

[revised manuscript text omitted]